# *Bacillus mycoides* PM35 Reinforces Photosynthetic Efficiency, Antioxidant Defense, Expression of Stress-Responsive Genes, and Ameliorates the Effects of Salinity Stress in Maize

**DOI:** 10.3390/life12020219

**Published:** 2022-01-30

**Authors:** Baber Ali, Xiukang Wang, Muhammad Hamzah Saleem, Muhammad Atif Azeem, Muhammad Siddique Afridi, Mehwish Nadeem, Mehreen Ghazal, Tayyaba Batool, Ayesha Qayyum, Aishah Alatawi, Shafaqat Ali

**Affiliations:** 1Department of Plant Sciences, Quaid-i-Azam University, Islamabad 45320, Pakistan; baberali@bs.qau.edu.pk (B.A.); atifazeem321@gmail.com (M.A.A.); mehwishnadeem1234@gmail.com (M.N.); taibabatool1954@gmail.com (T.B.); ayeshaaqt31@gmail.com (A.Q.); 2College of Life Sciences, Yan’an University, Yan’an 716000, China; 3College of Plant Science and Technology, Huazhong Agricultural University, Wuhan 430070, China; saleemhamza312@webmail.hzau.edu.cn; 4Department of Plant Pathology, Federal University of Lavras (UFLA), Lavras 37200-900, Brazil; msiddiqueafridi@gmail.com; 5Department of Botany, Bacha Khan University, Charsadda 24420, Pakistan; khanpyarieyes11@gmail.com; 6Biology Department, Faculty of Science, University of Tabuk, Tabuk 71421, Saudi Arabia; Amm.alatawi@ut.edu.sa; 7Department of Environmental Sciences, Government College University, Faisalabad 38000, Pakistan; 8Department of Biological Sciences and Technology, China Medical University, Taichung 40402, Taiwan

**Keywords:** abiotic stress, plant growth-promoting bacteria, plant–microbe interactions, salinity stress, bio-surfactant

## Abstract

Soil salinity is one of the abiotic constraints that imbalance nutrient acquisition, hampers plant growth, and leads to potential loss in agricultural productivity. Salt-tolerant plant growth-promoting rhizobacteria (PGPR) can alleviate the adverse impacts of salt stress by mediating molecular, biochemical, and physiological status. In the present study, the bacterium *Bacillus mycoides* PM35 showed resistance up to 3 M NaCl stress and exhibited plant growth-promoting features. Under salinity stress, the halo-tolerant bacterium *B. mycoides* PM35 showed significant plant growth-promoting traits, such as the production of indole acetic acid, siderophore, ACC deaminase, and exopolysaccharides. Inoculation of *B. mycoides* PM35 alleviated salt stress in plants and enhanced shoot and root length under salinity stress (0, 300, 600, and 900 mM). The *B. mycoides* PM35 alleviated salinity stress by enhancing the photosynthetic pigments, carotenoids, radical scavenging capacity, soluble sugars, and protein content in inoculated maize plants compared to non-inoculated plants. In addition, *B. mycoides* PM35 significantly boosted antioxidant activities, relative water content, flavonoid, phenolic content, and osmolytes while reducing electrolyte leakage, H_2_O_2_, and MDA in maize compared to control plants. Genes conferring abiotic stress tolerance (*CzcD, sfp,* and *srfAA* genes) were amplified in *B. mycoides* PM35. Moreover, all reactions are accompanied by the upregulation of stress-related genes (APX and SOD). Our study reveals that *B. mycoides* PM35 is capable of promoting plant growth and increasing agricultural productivity.

## 1. Introduction

Salinity stress negatively affects plant growth and development by unbalancing the nutritional and osmotic potential [1]. Salinity reduces approximately 25% of the agricultural yield in Pakistan [2]. About 50% of irrigated land and 20% of cultivated land have been affected by salt stress worldwide [3]. Soil salinization is caused by excessive irrigation, disposal of heavily salted groundwater, low precipitation, and high evaporation in dry regions [4]. Salinity stress is defined as the accumulation of excessive soluble salts in soil, which eventually have detrimental impacts on plant growth and development [5].

Plant morphology, physiology, biochemistry, and molecular processes are affected by salinity stress [6]. Seed germination and plant development are negatively affected by the osmotic and ionic imbalances of Na^+^ and Cl^−^ [7]. Specifically, salt stress causes the instability of cell structures, loss of membrane permeability, metabolic toxicity, and reactive oxygen species (ROS) production, which damages the cell ultrastructure and functions [8]. Salinity stress significantly decreases photosynthesis, chlorophyll content, leaf area, stomatal conductance, and the efficiency of photosystem II, which leads to growth hamper and productivity loss [9]. Moreover, the active uptake of nutrients decreases in plants growing under saline conditions [10]. It is crucial to develop and adapt various techniques to use saline land for agricultural productivity [11].

Microbial communities are natural technologies that could be exploited for plant growth and nutrient availability, with higher osmolyte accumulation (proline and sugars) under salinity stress conditions [12]. Among these microorganisms, plant growth-promoting rhizobacteria (PGPR) improve plant growth under salt stress conditions [13]. These PGPBs enhance plant growth and reduce salt absorption, which increases yield [14]. Under salt stress, the uptake and accumulation of plant nutrients such as N [15], P [16], and K reduces in plants. However, salt-tolerant plant growth-promoting rhizobacteria improve nutrient uptake and translocation in plants by employing biochemical and physiological mechanisms [17].

Salt-tolerant plant growth-promoting rhizobacteria are alternatives to chemical fertilizers as they play a pivotal role in nutrient acquisition by plants in the rhizosphere [18]. In this aspect, these bacteria are involved in the microbial synthesis of phytohormones such as indole-3-acetic acid (IAA), ethylene, cytokinins, and gibberellins. Furthermore, they can fix nitrogen and produce ACC deaminase, thus decreasing plant ethylene levels since ACC is an ethylene precursor. Besides these, plants acquire nutrients via phosphate solubilization and siderophore production by the plant growth-promoting rhizobacteria [19]. Most importantly, bacteria also produce exopolysaccharides (EPS), which are crucial in bacterial stress resistance, particularly under salt stress [20].

In Pakistan, maize (*Zea mays* L.) is the third most widely cultivated crop in the world and the third leading cereal crop after wheat and rice. Maize is also reported to be a moderately sensitive crop to salt [21]. Pakistan is the eighth-largest country to have salinity-affected land (6174.5 thousand acres) [22]. Thus, eco-friendly and sustainable strategies are employed to underpin sustainable agriculture in salt stress-affected regions across the globe. Plant growth-promoting bacteria (PGPB) produce phytohormones, proline, antioxidant enzymes, and possess stress-related genes. They can play a potential role in alleviating salinity stress and its adverse effects on maize physiology and biochemistry through direct and indirect mechanisms.

In developing countries, the human population is increasing exponentially, and cultivable land is constantly reducing. So, there is a need to produce abiotic-resistant varieties with high yield potential [23]. This study aimed to characterize *B. mycoides* PM35 based on its salinity tolerance, plant growth-promoting traits, the activity of extracellular enzymes, and its effects on maize growth under various salinity stress environments. The in-vitro and pot experiments revealed that *B. mycoides* PM35 is a salt-tolerant plant growth-promoting rhizobacteria that could potentially improve plant growth and alleviate salt stress by regulating molecular and biochemical mechanisms in salt-affected lands.

## 2. Materials and Methods

### 2.1. Procurement of Bacterial Strain

The bacterium *B. mycoides* PM35, obtained from Plant-Microbe Interactions Lab, Quaid-i-Azam University, Islamabad, Pakistan, was evaluated for salt tolerance potential at different concentrations of NaCl (0, 1, 2, and 3 M) [24]. *B. mycoides* PM35 tolerated up to 3 M NaCl and showed significant growth at all provided concentrations.

### 2.2. Salinity Tolerance Characteristics of B. mycoides PM35

#### 2.2.1. Bacterial Survivability

*B. mycoides* PM35 was cultured on LB medium amended with various concentrations of salinity stress [25]. Bacterial culture (20 µL) was inoculated in a TSB medium with NaCl (0, 300, 600, and 900 mM) and allowed to agitate continuously for 24 h. Total plate count (TPC) and serial dilution methods were applied to determine the bacterial population.

#### 2.2.2. Bacterial Flocculation

Bacterial strain *B. mycoides* PM35 was grown in TSB medium with 0, 300, 600, and 900 mM NaCl for 72 h at 30 °C, and flocculation was collected using Whatman No. 1 filter paper. It was oven-dried at 60 °C then after 2 h, the dry weight of the floc yield was measured [26].

#### 2.2.3. Bacterial Sodium Absorption

Bacterial strain *B. mycoides* PM35 was grown overnight at 30 °C in a TSB medium containing different NaCl concentrations (0, 300, 600, and 900 mM) and then centrifuged. Pellets were rinsed in sterile distilled water and digested in 0.1 N HCl at room temperature overnight. A flame photometer assessed bacterial sodium absorption [27].

#### 2.2.4. Biofilm Formation

We followed the protocol of Guimarães et al. [28] to quantify biofilm formation and used a microtiter plate-based technique with slight adjustments. *B. mycoides* PM35 was cultivated at 30 °C for 24 h on a salt-amended TSB medium (0, 300, 600, and 900 mM). After maintaining the optical density of bacterial cells at 0.3, 200 µL of bacterial cells were shifted in 96-well microtiter plate and incubated at 30 °C for 72 h. Biofilm present on walls of microtiter plate was stained with 0.01% crystal violet for 20 min after the evacuation of growth media. For quantification, biofilm was dissolved and analyzed at 590 nm. Then stained biofilm was extracted in 200 µL of 95% ethanol.

### 2.3. Quantitative Assays for Plant Growth-Promoting Traits

Indole-3-acetic acid (IAA) was quantitatively estimated using the colorimetric method [29]. Bacterial strain, *B. mycoides* PM35 was cultured in 50 mL LB-media supplemented with NaCl (0, 300, 600, and 900 mM), peptone, glycerol 15, and L-tryptophan (1 mg/mL). The culture was incubated in a shaker (160 rpm) at 30 °C for 7 d, centrifuged at 1000 rpm for 12 min, 1 mL of Salkowski reagent was added in 2 mL of supernatant and incubated for 30 min in the dark. The optical density was taken at 530 nm and compared with the standard curve. The standard curve of IAA (Sigma, St. Louis, MO, USA) was in the range of 10–100 µg/mL to estimate IAA concentration.

For determining siderophore production, Mehmood et al. was followed [30]. The bacterial culture was grown in a TSB medium under NaCl stress and centrifuged at 12,800 rpm for 10 min to collect the supernatant. The supernatant (0.5 mL) was mixed with equal amount of chrome azurol S reagent [CAS 121 mg/100 mL, 1 mM FeCl_3_ (20 mL) and HDTMA (20 mL)] and incubated for 20 min. Optical density was observed at 630 nm and siderophore, as percent siderophore unit (PSU) was estimated using the following formula:(1)$$\mathrm{PSU}\;=\mathrm{Ar}-\frac{\mathrm{As}}{\mathrm{Ar}}\times 100$$
where, As is inoculated sample absorbance and Ar is a reference (un-inoculated broth + CAS reagent + salt conc.).

Production of ACC deaminase enzyme by *B. mycoides* PM35 was determined using the protocol of Zainab et al. [31]. To quantify ACC deaminase production by bacterial cultures, they were grown in tryptic soy broth medium (TSB) for 24–48 h. The bacterial cells were harvested and centrifuged then pellets were washed with 0.1 M Tris HCl (pH = 7.5). The washed pellets were suspended in 2 mL of DF media containing 3 mM ACC supplemented with salinity stress (0, 300, 600, and 900 mM) and incubated the cultures for 24–48 h again at 32 °C. The bacterial cells were collected after 48 h by centrifugation at 300 rpm for 5 min, and pellets were washed with 2 mL of 0.1 M Tris HCl (pH = 7.5) and resuspended in 200 µL of 0.1 M Tris HCl (pH = 8.5). The bacterial pellets were labeled by adding 5% (*v*/*v*) toluene and then vortexed for 30 s. Then 50 µL of each sample was incubated with 5 µL of 0.3 M ACC at 28 °C for 30 min. Negative control had 50 µL of toluene-labeled cells without ACC. Blank included 50 µL of toluene-labeled cells with 0.3 M ACC. Samples were mixed with 500 µL of 0.56 M HCl and centrifuged at 12,000 rpm for 5 min. Each 500 µL sample was taken from negative and blank in a glass test tube and added 400 µL of 0.56 N HCl followed by 150 µL of 0.2% DNF solution and incubated for 30 min at 28 °C. Before taking absorbance at 540 nm, 1 mL of 2 N NaOH was added. The activity was estimated by the hydrolysis of ACC into α- ketobutyrate. A standard curve of α- ketobutyrate was drawn, ranging from 10–200 μmol and compared with absorbance taken at 540 nm of sample to determine μmol of α-ketobutyrate produced.

Exopolysaccharide’s production was tested under salinity stress by following the method of Zainab et al. [32] Bacterial strain *B. mycoides* PM35 was cultured in 50 mL of ATCC No. 14 media amended with salinity stress (0, 300, 600, and 900 mM NaCl) and incubated for 24–48 h at 32 °C and 150 rpm. The bacterial culture was centrifuged after 72 h for 20 min at 10,000 rpm. Acetone was added to the pellet and kept overnight at 40 °C. After 24 h, the pellet was dried at 100 °C and weighed. The amount of EPS produced was estimated as mg/mL of the dried weight.

### 2.4. Soil Collection, Analysis, and Seed Inoculation

The soil collected from the Quaid-i-Azam University, Islamabad, Pakistan (33.7470° N, 73.1371° E) was first air-dried in the laboratory, crushed, and sieved using a 2 mm sieve and sterilized by autoclaving at 121 °C for 30–40 min to avoid all microbes and fungal spores [24]. Physico-chemical properties were determined, including soil electrical conductivity, pH, organic matter, soil texture, available phosphorus, and potassium.

Certified maize seeds (SG-2002 Variety) collected from National Agricultural Research Center (NARC), Pakistan, were disinfected by serial washing with 70% ethyl alcohol for 5.0 min and 0.1% HgCl_2_ for 1.0 min. After disinfection, all seeds were three-time rinsed in autoclaved distilled water. Bacterial strain *B. mycoides* PM35 was cultured in 250 mL flasks containing LB broth. After 48 h, culture was taken and centrifuged for 10 min at 10,000 rpm to collect the pellet. The pellet was washed with 0.85% NaCl and resuspended in de-ionized water and, absorbance was adjusted to 0.5 to obtain a homogenous bacterial population (10^8^ CFU/mL) for inoculation. Seeds were dipped in bacterial solution for 2–4 h while uninoculated seeds soaked in sterilized water were considered control [33].

### 2.5. Pot Experiment under Controlled Conditions

The seeds (SG-2002 Maize variety) were sown (6 surface-sterilized seeds per pot) in plastic pots containing 200 g of sterilized soil. The concentration of salt stress in the pots was maintained (0, 300, 600, and 900 mM). In total, eight treatments were designed (in triplicate) in a complete randomized design (CRD) (Appendix A). All pots were placed in a growth chamber (CU-36L6, Perry, Iowa, US for 21 days. In each experimental unit, 20 mL of bacterial suspension (CFU/mL = 10^8^) was added. Pots were irrigated with 50 mL of distilled water daily to maintain moisture for plant growth. All the treatments were designed in triplicates. Throughout the experiment, EC and the pH of the substrate in each pot were kept constant. The same quantity of water was sprayed regularly to maintain 60–70% water holding capacity and balance NaCl levels in each pot. Humidity was maintained up to 60–80% in the growth chamber, the light duration for day and night was 12 h, and temperature range was 32 °C and 20 °C for day and night, respectively.

After 21 d of pot experiment, *Zea mays* L. plants were carefully harvested, and plant roots were washed vigorously under running tap water to remove soil particles from the root surface. After removing soil from the roots of plants, they were kept in bags and brought to the lab for further analyses.

### 2.6. Estimation of Agro-Morphological Parameters of Zea mays *L*.

Agro-morphological parameters, including plant height, length of shoot and root, and fresh and dry biomass, were analyzed for three plants selected randomly from each pot in all treatments and control after 21 d of cultivation. Plants were placed in an 80 °C hot air oven for 24 h to assess dry weight. The total leaf area was estimated and given in cm^2^/plant using the formula L× B × K, where L and B are the length and width of the leaves, respectively, and K is the Kemp’s constant (for Monocot 0.9) [34].

### 2.7. Estimation of Photosynthetic Pigments of Plants

Photosynthetic pigments of plants were estimated by following El-Esawi et al. [35] and using the following formula: Photosynthetic pigments were extracted by homogenizing 0.1 g of fresh leaves, with 6 mL of 80% ethanol. After centrifugation of the extract, the resulting supernatant was added to test tubes. The optical density of chlorophyll a, b, and carotenoids at 663, 645, and 470 nm was measured using a spectrophotometer (752 (N) UV-VIS, Beijing, China).
(2)Chlorophyll a =(12.7×A663)−(2.49×A645)
(3)Chlorophyll b =(12.9×A645)−(4.7×A663)
(4)Total chlorophyll =Chl a +Chl b
(5)Carotenoids =[(7.6×OD480)−1.49 (OD510)] × [(Final volume of filtrate/1000) × 0.5)]

### 2.8. Radical Scavenging Capacity of Leaves

Radical scavenging activity of the extracts was evaluated [36]. Fresh leaves of 100 mg were crushed in 80% methanol, centrifuged at 10,000 rpm, and the supernatant was collected. A suitable volume of supernatant (2 mL) and 180 µL of DPPH (Aldrich Chemistry, Burlington, VT, USA) solution (0.1 mM) were mixed. After 30 min, the mixture was colorless, and optical density (OD) was measured using a spectrophotometer (752 (N) UV-VIS, Beijing, China) at 517 nm.
(6)$$I\;\left(\%\right)=\;\mathrm{Ac}-\frac{\mathrm{As}}{\mathrm{Ac}}\times 100$$
where Ac = Control; As = Sample’s absorbance

### 2.9. Total Soluble Sugars (TSS)

Total soluble sugars (TSS) were determined by following Grad et al. [37]. Fresh leaves (0.1 g) were homogenized with 3–5 mL 80% ethanol to eliminate all traces of soluble sugars and centrifuged for 10 min at 10,000 rpm. The supernatant was collected and processed to calculate TSS. About 3 mL of freshly prepared anthrone solution and 0.1 mL of alcoholic extract were mixed in test tubes. All test tubes were heated for 12 min in boiling water and then iced for 10 min before being incubated for 20 min at 25 °C. The optical density of the solution was measured at 625 nm using a spectrophotometer (752 (N) UV-VIS, Beijing, China). The TSS was estimated in µg/mL of fresh weight using the glucose standard curve.

### 2.10. Protein Content of Leaves

The protein content of leaves was assessed in fresh leaves of maize using bovine serum albumin (BSA) (Sigma-Aldrich, St. Louis, MO, US as a reference according to the described protocol of Mendez and Kwon [38]. Fresh leaves (0.1 g) were crushed in a mortar and pestle with 1 mL of phosphate buffer (pH 7.5) and centrifuged for 10 min at 3000 rpm. The total volume of supernatant (0.1 mL) in test tubes was increased to 1 mL adding distilled water. Reagent C (Solution A and B in 50:1 ratio) (Solution A: 2% Na_2_CO_3_, 1% Na-K, 0.4% 0.1 N NaOH; Solution B: 0.5% CuSO_4_.5H_2_O in dH_2_O) (1 mL) was added, mixed for 10 min and then 0.1 mL of reagent D (Folin phenol: distilled water in a 1:1 ratio) was added. Different concentrations (20, 40, 60, 80, 320, and 640 mg) of the BSA solution were prepared then the absorbance of all samples was measured at 650 nm after 30 min of incubation.

### 2.11. Antioxidant Enzymatic Assays

Antioxidant activities in fresh leaves such as ascorbate peroxidase (APX), peroxidase (POD), and superoxide dismutase (SOD) were assessed, following the methodology of El-Esawi et al. [35] and Afridi et al. [39].

APX was analyzed using fresh leaf samples (0.2 g), initially crushed in 2 mL extraction buffer [potassium phosphate (PB), pH 7.5] and ascorbic acid (1 mM). The crushed samples were centrifuged at 15,000 rpm for 20 min at 4 °C, and optical density was calculated at 290 nm to assess APX.

For POD estimation, about 20 g of freshly collected plant tissues were crushed with 3 mL of 100 mM PB using a precooled pestle and mortar. The resulting homogenate was centrifuged at 10,000 rpm for 15 min at 4 °C, and optical density was calculated at 470 nm to assess POD.

The SOD activity of crushed plant material was assayed in 4 mL of solution (1 g PVP, 0.0278 g Na_2_EDTA), centrifuged at 10,000 rpm, the supernatant was collected, and then the volume was increased to 8 mL with PB (pH = 7.0). The absorbance at 560 nm was measured using a spectrophotometer. The SOD of plant samples was measured in units per 100 g FW.

The content of ascorbic acid (AA) in fresh leaves was determined using the protocol prescribed by Mohi-Ud-Din et al. [40]. The result was derived using an ascorbic acid (Sigma-Aldrich, St. Louis, Missouri, US) standard curve and expressed as nmoL/g FW.

### 2.12. Relative Water Content (RWC)

The relative water content (RWC) of green leaves was calculated by determining the turgid weight of fresh leaf samples and drying them in a hot air oven until they reached a consistent weight [33]. A 0.5 g (FW) leaf was placed in a petri dish, filled with distilled water, and left overnight in the dark. The turgid weight (TW) of leaves was calculated. The dry weight (DW) of the leaves was calculated by baking overnight at 72 °C.
(7)RWC (%)= ]FW−DW/TW−DW]×100

FW = Fresh weight; TW = Turgid weight; DW = Dry weight.

### 2.13. Total Flavonoids Content (TFC)

The aluminum chloride colorimetric technique, revised from Woisky and Salatino’s method [41], was used for determining total flavonoid content. Fresh leaves (100 mg) were homogenized in 3 mL of 80% methanol for TFC estimation [42]. In each tube, 0.50 mL of the extract was mixed with 1.50 mL of 95% ethanol and 0.10 mL of 10% AlCl_3_. At room temperature, the absorbance was measured at 415 nm with a UV-Vis spectrophotometer (UV-9200, Beijing, China) after 30 min of incubation. The calibration curve was generated using quercetin (Sigma, St. Louis, MO, USA). The quantitative evaluation was performed using a calibration curve with quercetin 1:1 (*w*/*v*) dissolved in absolute methanol as a reference and results were computed in milligrams of quercetin equivalents per 100 g fresh mass (mg QE/100). Each analysis was conducted thrice [43].

### 2.14. Total Phenolic Content (TPC)

Leaves were crushed and homogenized with 5 mL of 70% (*v*/*v*) methanol. After incubating samples for 30 min at 4 °C, they were centrifuged (at 15,000 rpm for 10 min), and resulting supernatants were analyzed [44]. Total phenolic content was measured spectrophotometrically using a technique based on Folin-phenolic Ciocalteu’s reagent (Merck, Taufkirchen, Germany) [45]. Folin-reagent (0.5 mL) and 0.45 mL of 7.5% (*w*/*v*) saturated sodium carbonate solution were added to methanol-extracted samples (20 µL). After a 2 h incubation period at 25 °C, samples’ absorbance was measured at 765 nm using a UV-VIS spectrophotometer (UV-9200, Beijing, China). Total phenolic compounds were computed and represented as mg gallic acid equivalent (mg GAE/100 g) sample using gallic acid (Sigma-Aldrich, St. Louis, MO, USA) as a reference (100–800 mg/L). The absorbance at 750 nm of the reaction mixture was measured spectrophotometrically.

### 2.15. Oxidative Stress Markers

The electrolyte leakage from leaf discs was measured using Sairam’s technique to calculate membrane stability index [46]. Leaf discs (0.10 g) from all treatments were placed in test tubes containing double distilled water. The EC of leaves was determined (C1) after 30 min in the water bath at 40 °C. The same leaf sample was subsequently maintained in a water bath at 100 °C for 10 min, and the EC was measured again (C2).
(8)Membrane Stability Index =[1−C1/C2]×100

The endogenous H_2_O_2_ content was determined using the modified method of Kapoor et al. [47]. Fresh weight (0.10 g) of leaf tissues was extracted with 3 mL of 0.1% trichloroacetic acid (TCA) in an ice bath and centrifuged at 12,000 rpm for 15 min to determine H_2_O_2_ concentration. One M potassium iodide (1.0 mL) and 10 mM potassium phosphate (0.50 mL) buffer (pH 7.00) were added to the supernatant. The absorbance of the supernatant was measured (at 390 nm). On a standardized curve, the content of H_2_O_2_ was expressed.

Malondialdehyde (MDA) quantification was estimated, following the Tulkova method [48]. In a cooled mortar and pestle containing 2 mL of 1% (*w*/*v*) trichloroacetic acid (TCA), a fresh leaf sample (0.2 g) was crushed. After centrifugation for 10 min at 15,000 rpm, 2 mL of the supernatant was removed and 4 mL of 0.5% thiobarbituric acid (TBA) was added to it. The mixture was heated to 95 °C and then allowed to cool. The absorbance, at 532 and 600 nm, of all treated samples was determined. The quantity of TBA was calculated using the absorption coefficient of 1.55 mmol/cm.
(9)$$\mathrm{MDA}\;=\Delta \left(\mathrm{OD}532-\mathrm{OD}600\right)/1.56\times 10^{5}$$

### 2.16. Osmo-Protectants Content

The free amino acids were determined using the ninhydrin technique described by Shafiq et al. [49]. Dried samples (200 mg) were homogenized in 5 mL of 80% alcohol and warmed for 15 min in a water bath. After that, the extract was centrifuged for 20 min at 2000 rpm. In a water bath, a 0.20 mL sample of the reaction mixture was heated with 3.80 mL of ninhydrin reagent. The reaction mixture was cooled until it became purple-blue. At 570 nm, absorbance was measured. The standard curve was constructed using leucine amino acid and findings were reported in mg of amino acid per gram of dry tissue.

The content of glycine betaine (GB) was determined [50]. For glycine betaine quantification, the extract was made by homogenizing 500 mg of dried leaves with 5.0 mL distilled water and 0.05% toluene, placed for 24 h. The reaction mixture was filtered using 0.20 mm micropore filters before centrifuging for 5 min at 6000 rpm. Then, 1.0 mL of HCl (2 N) and 0.10 mL of KI were stirred well with 0.50 mL of this extract. The mixture was chilled for two hours and then violently agitated. Ice-cold water (2 mL) and 10 mL 1, 2-dichloroethane, or dichloromethane were carefully mixed with this extract. After removal of the upper aqueous layer, the bottom, pink-colored layer was used to record optical density at 365 nm. The GB content in µg/gm dry weight was estimated using the betaine hydrochloride standard curve.

For the measurement of proline content in shoots, the method of Parveen and Siddiqui [51] was used. Fresh shoot material (0.2 g) was crushed in 3 mL of 3% sulphosalicylic acid, stored at 5 °C overnight. The obtained suspension was centrifuged for 5 min at 3000 rpm. The supernatant (2 mL) was blended with an acidic ninhydrin reagent after centrifugation. This reagent was prepared by dissolving 1.25 g ninhydrin in 20 mL phosphoric acid (6 M) and 30 mL glacial acetic acid (1 M H_3_PO_4_ = 3 N H_3_PO_4_) with constant stirring. The reagent was kept stable for 24 h. The tubes carrying the contents were heated for 1 h in a water bath at 100 °C. After cooling, the mixture was extracted with 4 mL toluene in a separate funnel. At 520 nm, optical density was determined using toluene as a blank.
(10)Proline µg/g = K×DF×Absorbance/FW

K = 17.52; Dilution factor = 2; Fresh weight = 0.5 g.

### 2.17. Amplification of CzcD and Bio-Surfactant Producing Genes

PCR was used to amplify the *CzcD* gene (398 bp) that encodes resistance to heavy metals (zinc-cadmium) by using the following two oligonucleotide primers: forward primer: 5′-CAGGTCACTGACACGACCAT-3′ and reverse primer: 5′-CATGCTGATGAGATTGATGATC-3′ in *Bacillus mycoides* PM35. The annealing temperature of primers was 57 °C. Negative and positive controls were included in the reaction [52].

The bio-surfactant gene *sfp*, was amplified from genomic DNA with two oligonucleotide primers; forward primer: *sfp* F: 5′-ATGAAGATTTACGGAATTTA-3′ and, reverse primer: *sfp* R: 5′-TTATAAAAGCTCTTCGTACG-3′ using PCR technique [53]. The thermal cycler conditions were as follows: an initial denaturation cycle of 1 min at 94 °C, followed by 25 cycles of 1 min denaturation at 94 °C, 30 s annealing at 46 °C, 1 min extension at 72 °C, and a 10 min final extension at 72 °C [53].

Similarly, PCR was used to amplify the *srfAA* gene (268 bp) that encodes surfactin production by using two primers forward primer; F-5′-TCGGGACAGGAAGACATCAT-3′; reverse primer: R-5′-CCACTCAAACGGATAATCCTGA-3′ [54]. The annealing temperature of primers was 58–60 °C. In the Gel Doc system (Universal Hood II, Los Angeles, California US), predicted bands for all genes were detected.

### 2.18. Gene Expression Analysis of Antioxidant (APX and SOD) Genes

The expression level of antioxidant genes (APX and SOD) was quantified using quantitative real-time PCR (qRT-PCR) in the presence and absence of *B. mycoides* PM35 under salinity stress (0, 300, 600, and 900 mM NaCl). Total RNA was extracted from maize plants using the Qiagen RNeasy Plant Mini kit. The cDNA was synthesized from RNA using the Qiagen Reverse Transcription kit. PCR amplification conditions were as described by El-Esawi et al. [55]. Primers, previously designed for the 2 antioxidant genes, were used for amplification [56]. The expression level of *Actin*, as a housekeeping gene, was determined following the 2^−ΔΔCt^ method.

### 2.19. Statistical Analysis

All treatment data were computed, with mean values and standard errors. Data were analyzed using analysis of variance (ANOVA) and pairwise comparison among treatment means by LSD test at *p* = 0.05 using Statistix 8.1. Principal Component Analysis (PCA) and Pearson correlation analysis were applied using R software.

## 3. Results

### 3.1. Growth Curve Analysis of B. mycoides PM35

*B. mycoides* PM35 tolerated 0, 1, 2, and 3 M NaCl and showed significant growth at all provided concentrations. Maximum bacterial growth appeared at 1 M concentration of NaCl rather than at 2 and 3 M concentrations. Growth curve analysis revealed a log phase at the fourth day of incubation (Figure 1). Optical density was highest for controls without stress.

### 3.2. Salinity Tolerance Traits of B. mycoides PM35 under Salinity Stress

The number of colony-forming units (CFUs) number in the culture medium plate showed the bacterial population. With increasing salinity, the bacterial population progressively declined in number (Figure 2). There was a substantial drop in the population of *B. mycoides* PM35 in the presence of 900 mM NaCl, compared to control.

NaCl concentration led to a significant increase in bacterial flocculation yield (Figure 2). At 900 mM NaCl, *B. mycoides* PM35 showed a significantly higher flocculation yield.

The production of biofilms was correlated to the production of EPS, with maximum production observed at 300 mM NaCl with a progressive reduction at 600 mM and 900 mM NaCl (Figure 2).

Figure 2 depicts the bacterial sodium uptake at various NaCl concentrations. The sodium absorption by *B. mycoides* PM35 was considerably higher at 900 mM NaCl, with values of 13.40 meq/L.

### 3.3. Quantitative Assay for Plant Growth-Promoting Traits of Bacteria under Salinty Stress

The production of auxins, siderophore, ACCD, and EPS improved the root morphology and physiology, thus leading to improved water and food absorption under salinity stress. Plant growth-promoting traits showed the potential of *B. mycoides* PM35 under salt stress. Four treatments at 0, 300, 600, and 900 mM showed that IAA production was directly proportional to NaCl concentrations. The high concentration resulted in an increased production of IAA with a maximum of 29.39% production at 900 mM NaCl concentration rather than control. Siderophore (10%), ACC (69%), and EPS (15%) production at 900 mM NaCl showed a similar trend as compared to control (Figure 3).

### 3.4. Physio-Chemical Properties of Soil

Appendix A showed the physico-chemical properties of soil. The soil texture of both soils (pre-sowing and post-harvesting) was loamy, slightly alkaline, and had an electrical conductivity of 1.53 and 4.49 dS/m, respectively. Organic matter analyzed in pre-sowing was higher than in post-harvested soil, and saturation was higher in pre-sowing soil as compared to post-harvested soil. However, available phosphorus and potassium content were higher in pre-sowing soil than in post-harvested soil.

### 3.5. Agro-Morphological Traits of Zea mays *L*.

After sowing surface-sterilized seeds in autoclaved soil, eight treatments (Appendix A) of *Zea mays* L. were harvested after 21 days to observe agro-morphological parameters. Salinity causes a severe reduction in growth in plants due to osmotic imbalance, ROS production, and lower water and nutrient uptake. In the current investigation, all parameters (shoot/root length, plant height, fresh/dry weight, and leaf surface area) showed the potential of *B. mycoides* PM35 in promoting plant growth and development (Figure 4). At 900 mM NaCl, shoot length was maximum in *B. mycoides* PM35 inoculated (32%) treatment than un-inoculated control. Similarly, all observed parameters revealed an increase in plant growth and biomass with *B. mycoides* PM35 inoculation than with un-inoculated controls (Table 1).

### 3.6. Photosynthetic Pigments of Plants

Chlorophyll content is a crucial marker for assessing plant health and photosynthetic efficiency under salt stress. In this study, analysis of Chl a, Chl b, and total chlorophyll revealed that *B. mycoides* PM35 inoculated plants possessed more pigment content (Chl a: 30–40%; Chl b: 32–47%; Total chl: 29–43%) as compared to the un-inoculated control. Pigmented content decreased with increasing saline stress (Table 2). However, all inoculated treatments were observed, with high values of photosynthetic pigments, showing the ability of *B. mycoides* PM35 to increase pigment content in maize plants. Similarly, total chlorophyll and carotenoids (30–47%) were higher in inoculated maize plants than in un-inoculated treatments (Table 2).

### 3.7. Radical Scavenging Capacity of Leaves

DPPH radical scavenging capacity is the measure of non-enzymatic antioxidant activity. Radical scavenging capacity in *B. mycoides* PM35 inoculated plants exhibited more DPPH content than un-inoculated under salinity stress (0, 300, 600, and 900 mM). However, this increase was greater at 0 mM (20%) and 300 mM (18%) as compared to 600 and 900 mM NaCl (Table 2).

### 3.8. Antioxidant Enzymes Assays

Antioxidants can indicate a plant’s tolerance to different stresses. We analyzed the production of enzymatic antioxidants (APX, POD, and SOD) and non-enzymatic antioxidants (AA) under salinity stress. Enzymatic antioxidants increased while non-enzymatic antioxidants decreased, with increasing NaCl concentration up to 900 mM (Figure 5). Maize plant with *B. mycoides* PM35 exhibited more accumulation (APX: 7–14%; POD: 34–53%; SOD: 13–15%) of these enzymes at all provided concentrations (0, 300, 600, and 900 mM) as compared to un-inoculated control treatments. Ascorbic acid content decreased with increasing salinity stress, while after inoculation of *B. mycoides* PM35, AA concentration significantly increased (13–33%) compared to un-inoculated maize plants (Figure 5).

### 3.9. Relative Water Content, Flavonoids, and Phenolic Content

Salt stress reduces root hydraulic conductivity, resulting in a decrease in water flows from roots to shoots. Analyses of relative water content, flavonoids, and phenolic content under salinity stress showed a significant decrease in these parameters along with increasing NaCl concentrations (Table 3). The *B. mycoides* PM35 inoculated maize plants revealed an increase in all tested parameters under control conditions (0 mM). Higher relative water content (44%) was expressed at 900 mM and 600 mM (37%) NaCl than at 300 mM (27%) NaCl as compared to the un-inoculated maize plants (Table 3). Flavonoid content significantly increased (23%) at 600 mM NaCl compared to the uninoculated control. However, phenolic content was highest (38%) at 900 mM NaCl than control (Table 3).

### 3.10. Total Soluble Sugars (TSS) and Protein Content of Leaves

Under high salinity stress, plants accumulate soluble solutes to mitigate the adverse effects of salt stress and maintain homeostasis. Plants treated with 0, 300, 600, and 900 mM NaCl exhibited decreased TSS with increasing salt stress, while the production of protein content was directly proportional to increasing salinity stress. After inoculating maize plants with *B. mycoides* PM35, a significant increase in TSS and protein content was observed compared to the un-inoculated plants (Table 3).

### 3.11. Oxidative Stress Markers

Salinity is a highly damaging factor that limits the growth and productivity of plants, mainly through oxidative stress. Untreated maize plants and those treated with *B. mycoides* PM35 investigated, under salinity stress, for the production of oxidative stress markers, showed that increasing the salt concentration from 0 to 900 mM, electrolyte leakage (ELL), H_2_O_2_, and MDA contents increased (Table 4). While *B. mycoides* PM35 inoculated treatments resulted in a significant decrease in all parameters at all provided concentrations of NaCl. The highest decrement in values was observed in ELL (12%), H_2_O_2_ (20%), and MDA (11%) at 900 mM NaCl as compared to un-inoculated treatments (Table 4).

### 3.12. Osmo-Protectant Content

Salinity stress triggers the production of reactive oxygen species (ROS) and modulates plant growth and physiology. Free amino acids, glycine betaine, and proline content were tested in the *B. mycoides* PM35 inoculated and uninoculated maize plants at 300, 600, and 900 mM NaCl concentrations. The production of osmo-protectants increased with the increasing salt stress without bacterial inoculation (Table 4). Free amino acid contents increased in *B. mycoides* PM35 treated samples at 900 mM (21%) and 600 mM (30.21%) salt concentration compared to un-inoculated controls. Glycine betaine and proline content increased with the inoculation of *B. mycoides* PM35 (Table 4). The GB (43%) and proline content (11%) were higher at 300 mM NaCl compared to control (Table 4).

### 3.13. Amplification of CzcD, sfp, and srfAA Genes

Polymerase chain reaction mediated amplification of abiotic stress-related genes (*CzcD, sfp,* and *srfAA*) was performed by using the above-mentioned pair of primers and resulted in a sharp band of approximately 398, 675, and 268 base pairs (bp), respectively (Figure 6).

### 3.14. Gene Expression Analysis

Compared to the non-inoculated controls in *B. mycoides* PM35, inoculation upregulated two antioxidant genes (APX and SOD) (Figure 7). Furthermore, compared to the non-inoculated salt-stressed plants, *B. mycoides* PM35 inoculated salinity-stressed maize plants showed significantly greater expression levels of antioxidant genes (Figure 7).

### 3.15. Principal Component and Pearson Correlation Analysis

Principal component biplot analysis showed a positive correlation between different variables under salinity stress with the application of *B. mycoides* PM35. Significantly correlated were placed very close, and in the same quadrate. Variable plot analysis showed 95% variations (Dim_1_ = 60.6%; Dim_2_ = 34.4%) (Figure 8). Shoot length, dry weight, plant height, pigmented content, organic compatible solutes, relative water content, flavonoids, phenolic content, antioxidant enzymes, and osmo-protectants showed positive correlations, while the radical scavenging capacity of leaves and oxidative stress markers negatively correlated with all other variables.

The Pearson correlation applied between the antioxidants and biochemical traits with plant biomass showed a highly positive correlation of chlorophyll a, b, total chlorophyll, and carotenoids with SL, FW, and RL in maize plants (Figure 9). An increase in these attributes is directly correlated with plant yield and biomass (Figure 9). There was a strong positive correlation between total soluble sugar, relative water content, total phenolic content, chlorophyll a, b, total chlorophyll, and carotenoids with SL, FW, and RL. Similarly, POD, TP, GB, APX, SOD, FA, and proline increased with the increasing plant biomass (SL, RL, and FW), biochemical traits, chlorophyll *a*, *b*, total chlorophyll, and carotenoids. The DPPH, ELL, MDA, and H_2_O_2_ showed a strong, negative relationship with all plant biomass attributes. However, the antioxidants, radical scavenging capacity, superoxide dismutase (SOD), peroxidase (POD), ascorbate peroxidase (APX), ascorbic acid (AA), total phenolic content, total flavonoid content, total soluble sugars, total protein, RWC (relative water content), electrolyte leakage, hydrogen peroxide, malondialdehyde, free amino acids, and glycine betaine showed a strong negative correlation. Decreasing the antioxidants also reduced plant biomass under different treatments (Figure 9).

## 4. Discussion

Sustainable agriculture demands a high yield and production of crops with the mitigation of abiotic stresses. The PGPB are well reported for stress tolerance and plant growth promotion [18]. Microbes developed complex physio-chemical mechanisms to maintain their survival and multiplication in salinity stress [57]. Rhizobacteria can influence plant growth directly or indirectly through many mechanisms [58]. In the present investigation, the tolerance potential of *B. mycoides* PM35 was screened against salinity stress. This bacterial strain indicated a high tolerance towards salinity stress (Figure 1).

In our study, *B. mycoides* PM35 had a low population of bacteria at 900 mM of NaCl modified medium (Figure 2), which is parallel to the previous report [59]. *B. mycoides* PM35 produced flocculation yield and biofilm at a much higher rate (Figure 2). The biofilm formation might link with protection strategies under salinity stress and nutrient deficiency [60]. The bacterial strain was grown in a nutrient-rich LB media in our study and showed better results. So, the biofilm formation was also improved when salt stress was applied in LB media of varying concentrations of NaCl. Biofilm formation is reduced by fewer nutrients in the media [61]. The results of our study depict that nutrients play a major role in the medium and regulate biofilm production (Figure 2).

The bacteria strain, *B. mycoides* PM35, significantly produced IAA, siderophore, ACC deaminase, and exopolysaccharides under salinity stress (Figure 3). In the current study, *B. mycoides* PM35 converted tryptophan into IAA under saline conditions, which is crucial for plant growth. Previous studies also revealed that the production of IAA by PGPB enhanced plant growth in *Bacillus mycoides* A1, *B. tequilensis* A3, *B. thuringiensis*, *Enterobacter* sp., and *Bacillus* sp. [62]. In addition, it promotes seed germination, root elongation, and improves root hair, which promotes mineral uptake in crops [63].

Siderophore production by bacterial strains is an important characteristic that significantly enhances the growth and development of plants [64]. PGPB contributes to the mobility of iron in the rhizosphere and increases its availability for plants. In our results, *B. mycoides* PM35 showed a similar trend as IAA under high-stress conditions (Figure 3), and enhanced plant growth up to 102% in controlled conditions. In this connection, a previous report showed that siderophore-producing bacteria colonized potato roots, sugar beets, and radishes, and consequently enhanced plant growth up to 144% in field trials [65]. Several other reports suggested that siderophore production by rhizospheric microflora enhanced Fe uptake by plants and improved their growth attributes [66].

Production of ACC deaminase was one of the key mechanisms of PGPB to suppress ethylene synthesis in plants under biotic and abiotic stresses. The ACC is the main precursor for the synthesis of the ethylene hormone in plants [67]. In the present study, ACC deaminase producing *B. mycoides* PM35 hydrolyzed ACC into ammonia and α-ketobutyrate in roots and diverted pathways of ethylene production under salinity stress (Figure 3). The ACC-producing plant-associated microorganisms tolerate abiotic stresses by alleviating the negative effects of ethylene production [68]. Various investigations proved that ACCD-producing bacteria alleviates the negative effects of salt stress by lowering ethylene levels, and consequently, improving plant growth under stress [69].

Some rhizosphere bacteria produce EPS or surface polysaccharides. *B. mycoides* PM35 produced significant EPS content under salinity stress as compared to control conditions (Figure 3). Although the composition and amount of EPS produced by various ST-PGPB strains vary, a large amount of EPS is produced in unfavorable circumstances (Khan and Bano, 2019). In addition, inoculating plants with EPS-producing PGPB improved their K^+^, Na^+^, and Ca^2+^ uptake [70]. Qurashi and Sabri [59] reported that chickpea development, soil structure, and aggregation were enhanced by EPS-producing ST-PGPB *Halomonasvariabilis* (HT1) and *Planococcusrifietensis* (RT4).

Salinity causes reduce plant growth severely, and creates osmotic stress by lowering water and nutrient uptake. It increases cellular ionic concentration and damages cellular biochemistry [71]. In our study, maize inoculated with *B. mycoides* PM35 showed better growth prospects as compared to non-inoculated plants under salt stress (Figure 4). The strain, *B. mycoides* PM35, stimulated shoot and root length, plant height, fresh, and dry weight, and leaf surface area of maize plants as compared with control plants (Table 1). The physiological parameters (Chl a, Chl b, Total chl, and carotenoids) were enhanced by applying *B. mycoides* PM35 under salt stress (Table 2). The current investigation was in-line with a previous study showing the inoculation of PGPB stimulated agro-morphological traits of wheat plants in a pot experiment in saline soil [72]. Our results are also consistent with the previous study that inoculation of PGPB enhanced the synthesis of chlorophyll content and promoted the growth of radish plants [73]. Tomato plants inoculated with PGPB, *Achromobacterpiechaudii* ARV8, improved the photosynthetic activity of plants under salinity stress [74]. Enhanced growth of the bacterial inoculated plants may be due to IAA and ACC-deaminase production by PGPB under stress conditions [75]. Enhanced photosynthetic activity may be due to the production of EPS content by PGPB under salinity stress. Exopolysaccharides protect plant seedlings from desiccation under salinity stress [24].

Under high salinity, plants accumulate soluble solutes to mitigate the toxic effects of salt stress and maintain ionic balance in cells [76]. In the present investigation, total soluble sugars and protein content increased with the inoculation of *B. mycoides* PM35 under salinity stress (Table 2). Previously, Qu et al. [77] and Zhang et al. [78] reported that plants inoculated with PGPB, or rhizobia, enhance the production of TSS and protein content in plants under NaCl stress. Modulating soluble sugars under high salinity results in CO_2_ assimilation and expression of associated genes [79]. Under stress, enhanced protein levels could result from the induction of stress-related protein biosynthesis [80].

The overproduction of ROS under stress conditions is a normal phenomenon that may lead to cell damage. However, plants produce several enzymatic and non-enzymatic substances to overcome the damage caused by ROS under stress conditions [81]. In the present study, the inoculation of *B. mycoides* PM35 to maize plants markedly increased the enzymatic (APX, POD, and SOD) and non-enzymatic (ascorbic acid) antioxidants under salt stress (Figure 5). These antioxidant substances may alleviate H_2_O_2_ and oxidative damage in *B. mycoides* PM35 inoculated plants compared with the control. Hashem et al. [82] reported that antioxidant activities are enhanced with the application of PGPB. In addition, Nunkaew et al. [83] investigated that 5-aminolevulinic acid-producing bacteria reduced the generation of H_2_O_2_ and increased the antioxidant activities of APX, POD, and SOD in salt-stressed rice plants [84]. The increased level in antioxidant enzymatic activity indicated that PGPB induced an antioxidant defense system in maize plants, eliminated toxic free radicals, and enhanced salt tolerance. These findings were in line with Habib et al. [85], who reported PGPB inoculated in okra plants.

Relative water content is an important factor for showing how better plants adapt to saline conditions. Bacterial strain *B. mycoides* PM35 inoculated maize plants showed a higher RWC compared to the control (Table 3). Salt stress reduced root hydraulic conductance, resulting in decreased water flow and stomatal closure [86]. Lawlor [87] reported that photosynthesis and transpiration were reduced adversely by the reduction of RWC in plants. Moreover, in the current investigation, *B. mycoides* PM35 induced flavonoid and phenolic pathways in maize subjected to saline stress and improved plant tolerance against this stress (Table 3). Bahadur et al. [88] investigated whether inoculation of PGPB accumulates the phenolic content in pea plants to mitigate salt stress. While the high accumulation of flavonoids and phenolic content in ST-PGPB treated plants under salinity stress may assist in the inactivation of ROS and decomposed H_2_O_2_ to prevent oxidative stress [89].

In the present study, electrolyte leakage, H_2_O_2_, and MDA content in *B. mycoides* PM35 inoculated maize plants were markedly reduced (Table 4). The ST-PGPB may regulate membrane function by scavenging excessive ROS produced in plant cells. These findings are consistent with those of Han et al. [90], where PGPB-inoculated maize and white clover plants reduced oxidative stress markers in a saline environment.

Osmo-protectants such as free amino acids, glycine betaine, and proline content are produced during salinity stress [91]. These solutes regulate water potential of the leaf and protect plants from osmotic shock [92]. Present findings demonstrated the increment in free amino acids, glycine betaine, and proline content in maize plants under a saline environment with the inoculation of *B. mycoides* PM35 (Table 4). Zarea et al. [93] reported that wheat inoculated with PGPB maintained its normal growth under salt stress by accumulating osmo-protectants. These findings also followed previous work showing the increased yield of wheat crops by applying PGPB under salinity stress [94].

In the current investigation, the heavy metal resistant *CzcD* and bio-surfactant producing *sfp* and *srfAA* genes for *B. mycoides* PM35 were amplified (Figure 6). The *sfp* and *srfAA* genes are essential components of the synthesis of the peptide system and play a vital role in the regulation of surfactant biosynthesis gene expression [95]. Banks et al. [96] stated that surfactants lower the surface tension of water penetrating the soil profile and increase the saturated area of soil. Plant roots may locate water in a wider soil profile with the surfactant, resulting in improved vegetative and generative proliferation and increased water efficiency. Altogether, *B. mycoides* PM35 improved plant growth and salt tolerance in maize plants, and we suggest it as a bio-fertilizer and multi-stress tolerant substance for plants.

Furthermore, *B. mycoides* PM35 inoculation dramatically increased the expression of genes linked to salt tolerance and antioxidant enzyme-encoding genes (Figure 7). These findings are consistent with Elkelish et al. [97], who found that salt stress enhanced the expression of SOD and APX in chickpeas. Ji et al. [98] revealed that PGPB-inoculated rice seedlings had greater levels of antioxidant gene expression, which improved salt stress tolerance.

## 5. Conclusions

The present study concludes that inoculation of halo-tolerant *B. mycoides* PM35 containing ACC deaminase and producing EPS significantly alleviates salinity stress in maize plants by producing proline and antioxidant enzymes. Inoculation of *B. mycoides* PM35 is an effective approach to mitigate salinity stress. The bacterial strain *B. mycoides* PM35 was able to tolerate NaCl concentrations up to 3 M. The pot inoculation study with *B. mycoides* PM35 significantly enhanced the growth and biomass of maize plants compared to non-inoculated plants under salinity stress. The bacterial strain, *B. mycoides* PM35, played a pivotal role in alleviating salinity stress by synthesis of antioxidant enzymes, compatible solutes, flavonoids, phenolic content, and accumulation of osmo-protectants under salinity stress. This promising bacterial strain also reduces the levels of electrolyte leakage, H_2_O_2_, and MDA content to relieve maize plants from salinity stress. Moreover, molecular profiling and expression of stress-related genes of *B. mycoides* PM35 supported its role in plant growth promotion under salinity stress and multi-stress tolerance. This current investigation in a pot experiment with inoculation of *B. mycoides* PM35 at different NaCl concentrations provides a baseline to analyze its potential under salinity stress. However, further research is required in natural saline field conditions with *B. mycoides* PM35 to validate its effectiveness and recommended its large-scale application in sustainable agriculture.

## Figures and Tables

**Figure 1 life-12-00219-f001:**
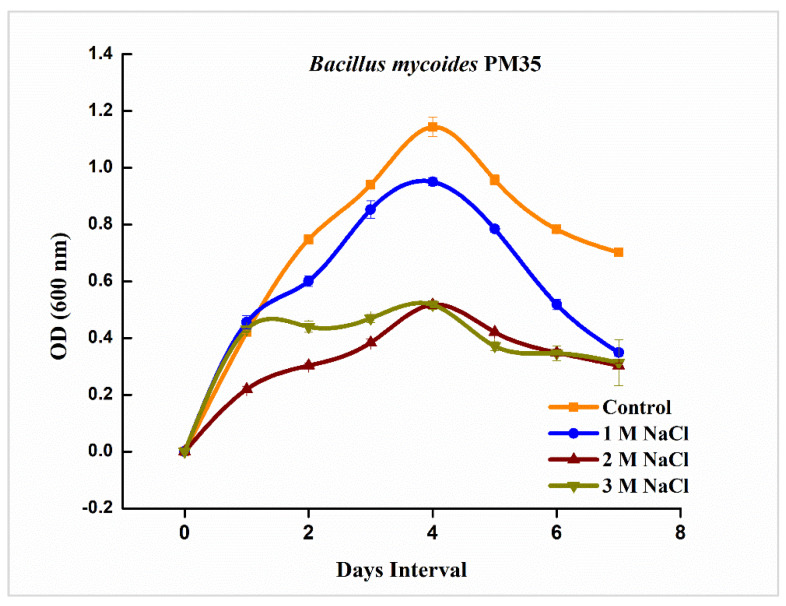
Growth curve analysis of *B. mycoides* PM35 under salinity stress (0, 1, 2 and 3 M NaCl).

**Figure 2 life-12-00219-f002:**
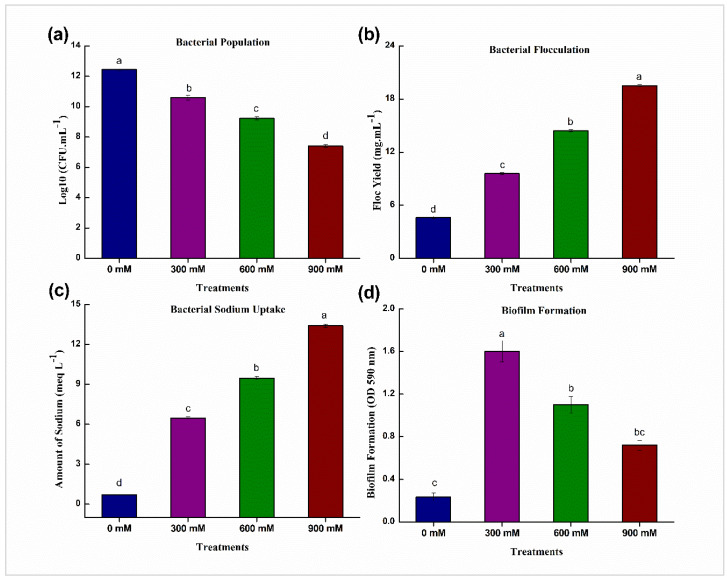
Effects of NaCl on salinity tolerance traits of *B. mycoides* PM35 (**a**) Bacterial Population (**b**) Flocculation Yield (**c**) Bacterial Sodium Uptake (**d**) Biofilm Formation. Bars sharing different letter(s) for each parameter are significantly different from each other according to Least Significant Difference (LSD) test (*p* ≤ 0.05). All the data represented are the average of three replications (*n* = 3). Error bars represent the standard errors (SE) of three replicates.

**Figure 3 life-12-00219-f003:**
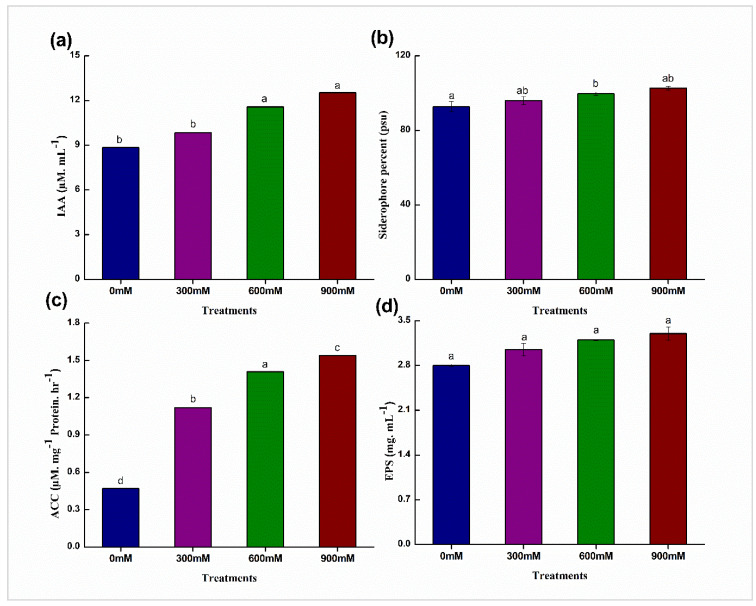
Quantitative estimation of the following PGP traits of *B. mycoides* PM35: (**a**) Indole-3- acetic acid (IAA) (**b**) Siderophore (**c**) ACC deaminase (ACCD) (**d**) Exopolysaccharides (EPS). Bars sharing different letter(s) for each parameter are significantly different from each other according to Least Significant Difference (LSD) test (*p* ≤ 0.05). All the data represented are the average of three replications (*n* = 3). Error bars represent the standard errors (SE) of three replicates.

**Figure 4 life-12-00219-f004:**
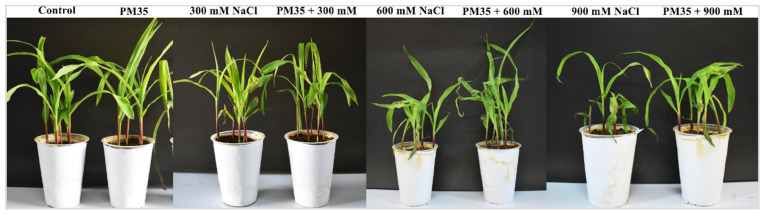
Effects of *B. mycoides* PM35 on plant growth promotion of *Zea mays* L. under salinity stress.

**Figure 5 life-12-00219-f005:**
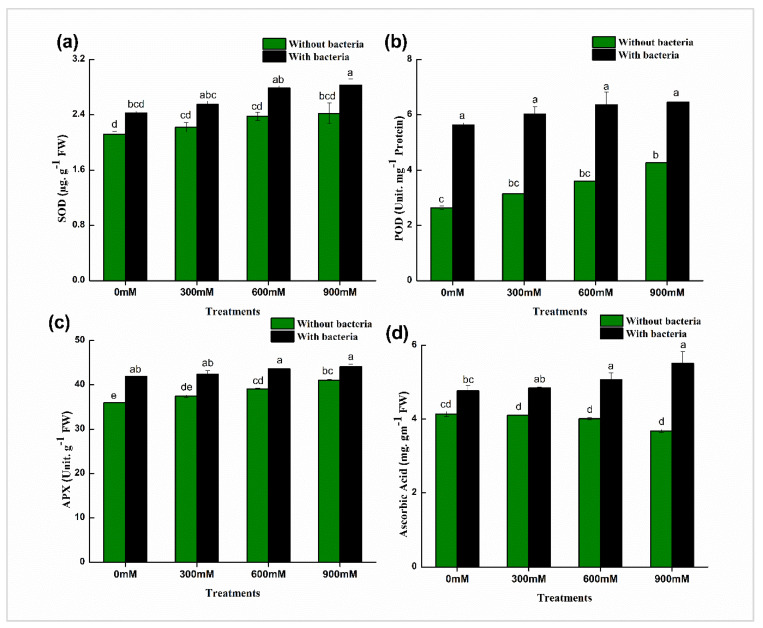
Effects of *B. mycoides* PM35 on levels of enzymatic and non-enzymatic antioxidants; (**a**) Superoxide dismutase (SOD) (**b**) Peroxidases (POD) (**c**) Ascorbate peroxidase (APX) (**d**) Ascorbic Acid. Bars sharing different letter(s) for each parameter are significantly different from each other according to Least Significant Difference (LSD) test (*p* ≤ 0.05). All the data represented are the average of three replications (*n* = 3). Error bars represent the standard errors (SE) of three replicates.

**Figure 6 life-12-00219-f006:**
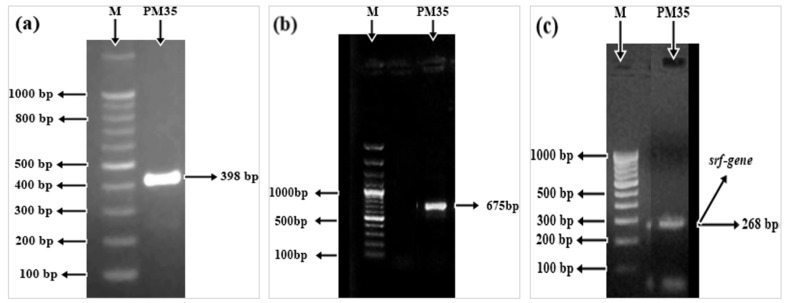
Amplification of abiotic stress-related genesin *B. mycoides* PM35: (**a**) *CzcD*-gene (**b**) *sfp*-gene (**c**) *srfAA*-gene.

**Figure 7 life-12-00219-f007:**
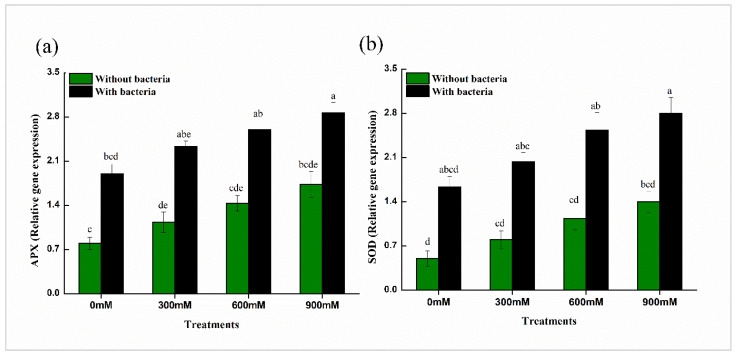
Expression levels of antioxidant genes of maize in the absence and presence of *B. mycoides* PM35 under salinity stress, (**a**) Ascorbate peroxidase (APX) (**b**) Superoxide dismutase (SOD). Bars sharing different letter(s) for each parameter are significantly different from each other according to Least Significant Difference (LSD) test (*p* ≤ 0.05). All the data represented are the average of three replications (*n* = 3). Error bars represent the standard errors (SE) of three replicates.

**Figure 8 life-12-00219-f008:**
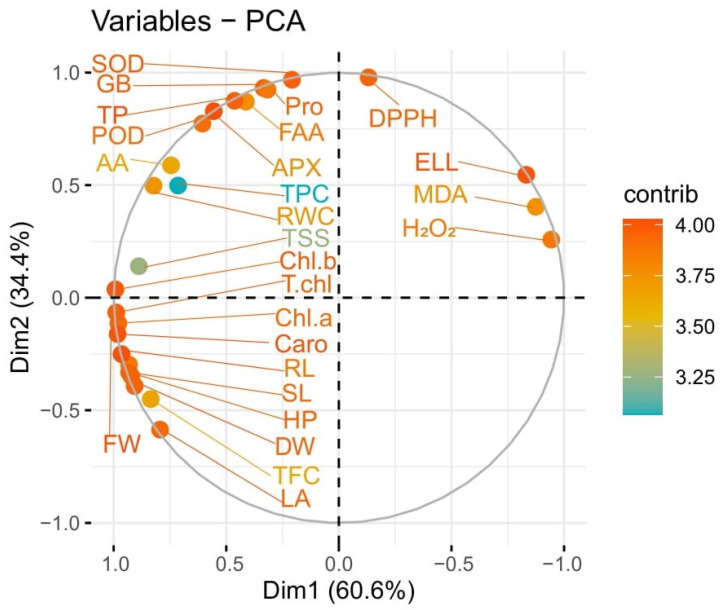
PCA biplot showing the categorization of *B. mycoides* PM35 based on its effects on maize growth-promoting characteristics under salinity stress.

**Figure 9 life-12-00219-f009:**
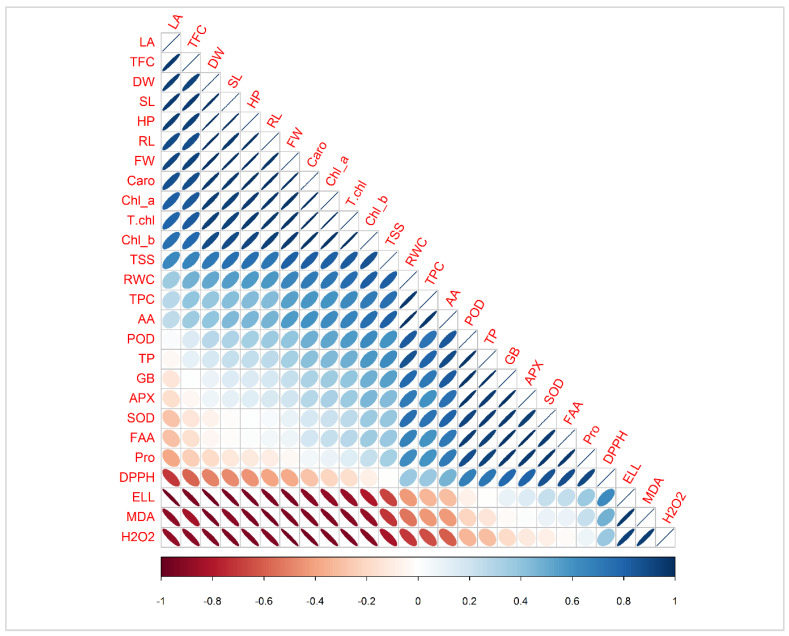
Pearson correlation between antioxidants and biochemical traits with plant biomass parameters under various salt stresses; Pro, (proline), SL (shoot length), RL (root length), PH (plant height), FW (fresh weight), DW (dry weight), LA (leaf area), Chl a (chlorophyll a), Chl b (chlorophyll b), T. chl (total chlorophyll), Caro (carotenoids), DPPH (radical scavenging capacity), SOD (superoxide dismutase), POD (peroxidase), APX (ascorbate peroxidase), AA (ascorbic acid), TPC (total phenolic content), TFC (total flavonoid content), TSS (total soluble sugars), TP (total protein), RWC (relative water content), ELL (electrolyte leakage), H_2_O_2_ (hydrogen peroxide), MDA (malondialdehyde), FAA (free amino acids), and GB (glycine betaine).

**Table 1 life-12-00219-t001:** Maize growth, biomass, and leaf surface area in the presence and absence of *B. mycoides* PM35 under salinity stress.

NaCl (mM)	*B. mycoides* PM35	SL (cm)	RL (cm)	PH (cm)	FW (g)	DW (g)	LA (cm^2^)
0 mM	−PM35	34.33 ± 0.93 ^a^	17 ± 0.53 ^bc^	51.33 ± 1.46 ^b^	2.17 ± 0.25 ^ab^	0.33 ± 0.02 ^abc^	17.95 ± 0.95 ^ab^
	+PM35	45.33 ± 1.07 ^b^	24.66 ± 0.93 ^a^	70 ± 0.26 ^a^	2.89 ± 0.08 ^a^	0.44 ± 0.02 ^a^	19.95 ± 0.93 ^a^
300 mM	−PM35	25 ± 1.21 ^cd^	13.66 ± 0.55 ^cd^	38.66 ± 1.53 ^cd^	1.68 ± 0.14 ^bc^	0.26 ± 0.01 ^bcd^	13.79 ± 0.94 ^bc^
	+PM35	34 ± 1.06 ^b^	20.66 ± 1.33 ^ab^	54.66 ± 2.01 ^b^	2.32 ± 0.11 ^ab^	0.35 ± 0.01 ^ab^	15.55 ± 0.99 ^abc^
600 mM	−PM35	21.33 ± 0.93 ^de^	11 ± 0.95 ^cd^	33.33 ± 1.86 ^de^	1.46 ± 0.10 ^bc^	0.22 ± 0.01 ^cd^	12.43 ± 0.50 ^bc^
	+PM35	30.33 ± 1.33 ^bc^	17.33 ± 1.60 ^bc^	47.66 ± 2.94 ^bc^	2.11 ± 0.13 ^ab^	0.30 ± 0.01 ^bc^	13.82 ± 0.84 ^bc^
900 mM	−PM35	18 ± 0.70 ^e^	8.00 ± 0.79 ^d^	26 ± 1.48 ^e^	1.08 ± 0.06 ^c^	0.18 ± 0.01 ^d^	10.19 ± 0.78 ^c^
	+PM35	26.33 ± 0.81 ^cd^	12.66 ± 1.19 ^cd^	39.33 ± 1.81 ^cd^	1.87 ± 0.13 ^bc^	0.25 ± 0.01 ^bcd^	12.94 ± 0.92 ^bc^

Growth was measured at 21 days after seed sowing under different salt concentration regimes. SL–Shoot length, RL–Root length, PH–Plant height, FW–Fresh weight, DW–Dry weight. The treatments exhibit dissimilar letters within rows, representing significance (*p* ≤ 0.05).

**Table 2 life-12-00219-t002:** Pigmented contents, DPPH activity, total soluble sugars, and proteins in presence and absence of *B. mycoides* PM35 under salinity stress.

NaCl (mM)	*B. mycoides* PM35	*B. mycoides* PM35	Chl a (mg/g FW)	Chl b (mg/FW)	Total Chl (mg/g FW)	Carotenoids (mg/g FW)	DPPH (IC_50_)%
0 mM	−PM35	15.18 ± 1.11 ^bc^	7.14 ± 0.40 ^bcde^	22.94 ± 0.82 ^cd^	7.1 ± 0.59 ^bc^	35.1 ± 1.37 ^e^
	+PM35	21.59 ± 0.53 ^a^	10.61 ± 0.30 ^a^	32.2 ± 0.83 ^a^	10.13 ± 0.40 ^a^	44.15 ± 1.07 ^f^
300 mM	−PM35	12.6 ± 0.77 ^cde^	6.38 ± 0.56 ^cde^	18.98 ± 0.89 ^def^	5.32 ± 0.30 ^cde^	49.45 ± 1.59 ^de^
	+PM35	20.22 ± 0.69 ^ab^	9.40 ± 0.55 ^ab^	29.62 ± 1.08 ^ab^	8.54 ± 0.38 ^ab^	60.18 ± 1.08 ^bc^
600 mM	−PM35	10.87 ± 0.38 ^de^	5.61 ± 0.43 ^de^	16.48 ± 0.39 ^ef^	4.42 ± 0.42 ^de^	55.07 ± 1.47 ^cd^
	+PM35	17.15 ± 0.66 ^abc^	8.75 ± 0.57 ^abc^	25.9 ± 1.19 ^bc^	7.43 ± 0.15 ^bc^	65.86 ± 1.08 ^ab^
900 mM	−PM35	8.75 ± 0.63 ^e^	4.26 ± 0.22 ^e^	13.01 ± 0.76 ^f^	3.34 ± 0.34 ^e^	62.81 ± 1.19 ^abc^
	+PM35	14.54 ± 0.79 ^cd^	7.99 ± 0.39 ^abcd^	22.53 ± 1.19 ^cde^	6.32 ± 0.33 ^bcd^	70.93 ± 0.97 ^a^

The chlorophyll and relative water content in leaves were measured after 21 days of seed sowing. Chl a–chlorophyll a, Chl b–chlorophyll, Total Chl–Total chlorophyll, and carotenoids, DPPH–2,2-diphenyl-1-picrylhydrazyl. The treatments exhibit dissimilar letters within rows represent significance (*p* ≤ 0.05).

**Table 3 life-12-00219-t003:** Relative water content, soluble solutes (total flavonoid and phenolic content), total soluble sugars, and proteins in the presence and absence of *B. mycoides* PM35 under salinity stress.

NaCl (mM)	*B. mycoides* PM35	RWC (%)	TFC (mg QE/g FW)	TPC (mg GAE/g)	Soluble Sugars (mg/g FW)	Proteins (mg/g FW)
0 mM	−PM35	52.03 ± 1.25 ^b^	117.93 ± 0.26 ^a^	11.51 ± 0.26 ^cd^	78.3 ± 0.02 ^c^	0.21 ± 0.01 ^d^
	+PM35	62.29 ± 1.21 ^c^	120.67 ± 1.07 ^b^	11.9 ± 0.02 ^bc^	90.73 ± 0.00 ^a^	0.47 ± 0.01 ^b^
300 mM	−PM35	47.86 ± 0.76 ^cd^	80.83 ± 0.62 ^e^	10.65 ± 0.06 ^de^	73.71 ± 0.03 ^e^	0.25 ± 0.01 ^cd^
	+PM35	65.06 ± 1.46 ^ab^	83.74 ± 1.15 ^c^	12.44 ± 0.02 ^b^	86.41 ± 0.00 ^b^	0.51 ± 0.01 ^b^
600 mM	−PM35	43.88 ± 1.06 ^d^	75.19 ± 0.16 ^f^	10.16 ± 0.01 ^e^	67.98 ± 0.01 ^g^	0.30 ± 0.01 ^cd^
	+PM35	69.17 ± 1.25 ^ab^	96.8 ± 0.26 ^c^	13.84 ± 0.02 ^a^	75.9 ± 0.00 ^d^	0.57 ± 0.02 ^ab^
900 mM	−PM35	40.56 ± 1.21 ^d^	71 ± 0.12 ^g^	8.99 ± 0.26 ^f^	63.84 ± 0.03 ^h^	0.35 ± 0.00 ^c^
	+PM35	71.91 ± 1.32 ^a^	87.77 ± 0.26 ^d^	14.5 ± 0.00 ^a^	91.01 ± 0.02 ^f^	0.67 ± 0.02 ^a^

RWC: Relative water content; TFC: Total flavonoid content; TPC: Total phenolic content; QE: Quercetin; GAE: Gallic acid; FW: Fresh weight. The treatments exhibit dissimilar letters within rows represent significance (*p* ≤ 0.05).

**Table 4 life-12-00219-t004:** Level of oxidative stress markers and osmoprotectants in the presence and absence of *B. mycoides* PM35 under salinity stress.

NaCl (mM)	*B. mycoides* PM35	ELL (%)	H_2_O_2_ (µmoL/g FW)	MDA (nmoL/g FW)	AA (mg/g DW)	GB (µg/g DW)	Proline (µg/g FW)
0 mM	−PM35	48.4 ± 0.03 ^f^	24.87 ± 0.40 ^bc^	5.65 ± 0.01 ^c^	7.3 ± 0.23 ^e^	3.31 ± 0.32 ^e^	57.18 ± 0.20 ^e^
	+PM35	44.6 ± 0.05 ^g^	22.8 ± 0.68 ^c^	5.2 ± 0.02 ^d^	13.06 ± 0.40 ^bc^	6.09 ± 0.27 ^bc^	64.55 ± 0.37 ^cd^
300 mM	−PM35	57 ± 0.27 ^d^	28.24 ± 0.68 ^b^	5.91 ± 0.03 ^c^	9.56 ± 0.29 ^d^	3.81 ± 0.22 ^e^	59.71 ± 0.12 ^e^
	+PM35	50.4 ± 0.21 ^e^	24.88 ± 0.40 ^bc^	5.31 ± 0.08 ^d^	15.81 ± 0.15 ^a^	6.56 ± 0.24 ^bc^	66.81 ± 0.34 ^bc^
600 mM	−PM35	64.6 ± 0.03 ^b^	31.72 ± 0.27 ^a^	6.42 ± 0.00 ^b^	11.41 ± 0.26 ^c^	4.48 ± 0.09 ^de^	63.61 ± 0.38 ^d^
	+PM35	57.6 ± 0.06 ^d^	26.06 ± 0.52 ^bc^	5.90 ± 0.04 ^c^	16.35 ± 0.25 ^a^	6.99 ± 0.02 ^ab^	69.49 ± 0.80 ^b^
900 mM	−PM35	70.1 ± 0.03 ^a^	34.03 ± 0.28 ^a^	7.06 ± 0.01 ^a^	13.75 ± 0.14 ^b^	5.45 ± 0.16 ^cd^	66.53 ± 0.33 ^bcd^
	+PM35	61.9 ± 0.04 ^c^	27.33 ± 0.71 ^bc^	6.27 ± 0.04 ^bee^	17.46 ± 0.27 ^a^	8.07 ± 0.04 ^a^	73.75 ± 0.64 ^a^

The effect of NaCl treatments under different salt concentration conditions. ELL–Electrolyte leakage, H_2_O_2_–Hydrogen peroxide, MDA–Malondialdehyde, AA–Amino Acid, GB–Glycine betaine and proline. The treatments exhibit dissimilar letters within rows represent significance (*p* ≤ 0.05).

## Data Availability

The paper reflects the authors’ own research and analysis in a truthful and complete manner. The paper is not currently being considered for publication elsewhere. All authors have been personally and actively involved in substantial work leading to the paper and will take public responsibility for its content.

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
