# Peer review of "Bacillus mycoides PM35 Reinforces Photosynthetic Efficiency, Antioxidant Defense, Expression of Stress-Responsive Genes, and Ameliorates the Effects of Salinity Stress in Maize"

_life, 2022, doi:10.3390/life12020219_

Round 1
Reviewer 1 Report
Using biochemical, physiological, and molecular tools, Ali et al., analyzed the effect of Bacillus mycoides PM35 strain on salt tolerance in maize plants. They found that inoculated maize plants with PM35 strain improve plant growth and development by regulating auxin and ethylene pathways, ROS production, osmolytes production, etc.
The results generated in this study are well documented and described. This work adds to the options to improve plant growth in response to multiple stressors, including salt tolerance. However, I have minor comments.
1.- Bacillus mycoides should be in italics. Correct it in the overall manuscript, including the title.
2.- The title should be modified, be more concrete.
3.- Change slat by salt in the third paragraph of the Introduction.
4.- In-vitro should be in italic. (Last paragraph in Introduction).
5.- To include in material and methods the protocol used for IAA and ACC determination.
6.- Authors should modify the format of “Y” axis. In several graphs. The Y axis, must star with 0 and not with 0.0.
7.- Authors should analyze the cell viability in roots of maize plants inoculated and un-inoculated with PM35 strain, since ROS production induced by salt stress, triggers cell damage in root tissues.
8.- Authors could analyze the ABA levels and/or ABA gene expression. It has been that salt tolerance involve ABA responses.
9.- In the Results section, authors should include a small introduction for each result. Also, the result section should be more connections between each result chapter.
Author Response
Author's Response to Reviewer Comments
Dear Editor,
The authors gratefully acknowledge the reviewers for spending valuable time reviewing our manuscript and making constructive comments and suggestions. We believe that having followed the reviewer’s comments, the scientific and technical quality of the paper has been improved and fulfills the publication requirements of your esteemed journal. All the revisions are included by using the “track changes function” in the “manuscript with track changes file”
We are looking forward to hearing from you.
Yours faithfully
Baber Ali
(On behalf of Co-authors)
Reviewer 1:
Open Review
(x) I would not like to sign my review report
( ) I would like to sign my review report
English language and style
( ) Extensive editing of English language and style required
( ) Moderate English changes required
( ) English language and style are fine/minor spell check required
(x) I don't feel qualified to judge about the English language and style
|
|
|
Can be improved |
Must be improved |
Not applicable |
|
Does the introduction provide sufficient background and include all relevant references? |
(x) |
( ) |
( ) |
( ) |
|
Is the research design appropriate? |
(x) |
( ) |
( ) |
( ) |
|
Are the methods adequately described? |
( ) |
(x) |
( ) |
( ) |
|
Are the results clearly presented? |
(x) |
( ) |
( ) |
( ) |
|
Are the conclusions supported by the results? |
(x) |
( ) |
( ) |
( ) |
First of all, thank you so much for taking the time to review our manuscript. I am sure your considerations and corrections will give more reading comprehension and add more value to the paper. On behalf of the other authors, I appreciate the feedback from the Reviewer.
All suggestions made by the Reviewer were taken into account and the responses are highlighted with yellow color.
Comments and Suggestions for Authors
Using biochemical, physiological, and molecular tools, Ali et al., analyzed the effect of Bacillus mycoides PM35 strain on salt tolerance in maize plants. They found that inoculated maize plants with PM35 strain improve plant growth and development by regulating auxin and ethylene pathways, ROS production, osmolytes production, etc.
The results generated in this study are well documented and described. This work adds to the options to improve plant growth in response to multiple stressors, including salt tolerance. However, I have minor comments.
- - Bacillus mycoides should be in italics. Correct it in the overall manuscript, including the title.
Response: Thanks for the comment and Bacillus mycoides has been changed into italic throughout the MS.
- - The title should be modified, be more concrete.
Response: Thank you for your feedback. The title of the MS has been modified from “Bacillus mycoides PM35 Induces Salt Tolerance in Maize by Regulating Redox Potential, Ethylene Production and Stress-related Gene Amplification under Salt Stress Regimes” to “Bacillus mycoidesPM35 Reinforces Photosynthetic Efficiency, Antioxidant Defense, Expression of Stress-Responsive Genes and Ameliorates the Effects of Salinity Stress in Maize”
- - Change slat by salt in the third paragraph of the Introduction.
Response: Thanks for the comment. Slat has been revised with “salt” in the third paragraph of the Introduction.
- - In-vitro should be in italic. (Last paragraph in Introduction).
Response: Well, In-vitro has been revised in italic format.
- - To include in material and methods the protocol used for IAA and ACC determination.
Response: Thanks, we agree with the reviewer and described the protocols in more detail in the material and methods section of MS.
- - Authors should modify the format of “Y” axis. In several graphs. The Y axis, must star with 0 and not with 0.0.
Response: Well, It's impossible to start Y-axis from the whole number when all the values on Y-axis are in points as it is automatically generated by the software.
- - Authors should analyze the cell viability in roots of maize plants inoculated and un-inoculated with PM35 strain, since ROS production induced by salt stress, triggers cell damage in root tissues.
Response: We understand your concern but at this stage, our research project has been completed. We acknowledge your precious suggestion and will definitely follow in our upcoming research projects.
- - Authors could analyze the ABA levels and/or ABA gene expression. It has been that salt tolerance involves ABA responses.
Response: We appreciate your kind suggestion to analyze the ABA levels and/or ABA gene expression but at this time, we mention above that our research project has been completed. We analyzed APX and SOD gene expression by taking into account our lab resources and funding in our laboratory.
- - In the Results section, authors should include a small introduction for each result. Also, the result section should be more connections between each result chapter.
Response: Thank you for your suggestion. We have revised the whole result section in MS and added an introduction for each result to develop more connections between each result.
Reviewer 2 Report
The article entitled “Bacillus mycoides PM35 Induces Salt Tolerance in Maize by Regulating Redox Potential, Ethylene Production and Stress-related Gene Amplification under Salt Stress Regimes" presents the results of a study which aimed to investigate the biochemical and molecular responses of Bacillus mycoides PM35 tolerance and its effects on maize growth under various salinity stress environments.
The subject of the study is very interesting and topical, with scientific and practical importance. The data presented are of more than national interest.
The introduction is presented correctly, in accordance with the subject. Numerous scientific articles, in concordance to the topic of the study, were consulted.
Methodology of the study was clearly presented, and appropriate to the proposed objectives.
The obtained results are important and have been analyzed and interpreted correctly, in accordance with the current methodology.
The discussions are appropriate, in the context of the results, and was conducted compared to other studies in the field.
The scientific literature, to which the reporting was made, is recent and representative in the field.
There are some minor changes I am suggesting in detailed comments below.
Detailed comments
- Introduction
Na+ and Cl- - wrong format
ROS - The abbreviation “ROS " appears in the work for the first time - it should be explained.
2.3. Quantitative assays for PGP traits
FeCl3 - wrong format. Check and correct throughout the manuscript
2.4. Soil collection, analysis and seed inoculation
(CFU) mL−1] - wrong format. Check and correct throughout the manuscript
2.8. Radical scavenging capacity of Leaves
OD - The abbreviation “OD” appears in the work for the first time - it should be explained.
Below the tables and figures, please add an explanation of what the lowercase letters in the columns mean
Figure 6 - Check and correct format
Author Response
Author's Response to Reviewer Comments
Dear Editor,
The authors gratefully acknowledge the reviewers for spending valuable time reviewing our manuscript and making constructive comments and suggestions. We believe that having followed the reviewer’s comments, the scientific and technical quality of the paper has been improved and fulfills the publication requirements of your esteemed journal. All the revisions are included by using the “track changes function” in the “manuscript with track changes file”
We are looking forward to hearing from you.
Yours faithfully
Baber Ali
(On behalf of Co-authors)
Reviewer 2:
Open Review
(x) I would not like to sign my review report
( ) I would like to sign my review report
English language and style
( ) Extensive editing of English language and style required
( ) Moderate English changes required
( ) English language and style are fine/minor spell check required
(x) I don't feel qualified to judge about the English language and style
|
|
|
Can be improved |
Must be improved |
Not applicable |
|
Does the introduction provide sufficient background and include all relevant references? |
(x) |
( ) |
( ) |
( ) |
|
Is the research design appropriate? |
(x) |
( ) |
( ) |
( ) |
|
Are the methods adequately described? |
(x) |
( ) |
( ) |
( ) |
|
Are the results clearly presented? |
(x) |
( ) |
( ) |
( ) |
|
Are the conclusions supported by the results? |
(x) |
( ) |
( ) |
( ) |
First of all, thank you so much for taking the time reviewing our manuscript. I am sure your considerations and corrections will give more reading comprehension and add more value to the paper. On behalf of the other authors, I appreciate the feedback from the Reviewer.
All suggestions made by the Reviewer were taken into account and the responses are highlighted with yellow color.
Comments and Suggestions for Authors
The article entitled “Bacillus mycoides PM35 Induces Salt Tolerance in Maize by Regulating Redox Potential, Ethylene Production and Stress-related Gene Amplification under Salt Stress Regimes" presents the results of a study which aimed to investigate the biochemical and molecular responses of Bacillus mycoides PM35 tolerance and its effects on maize growth under various salinity stress environments.
The subject of the study is very interesting and topical, with scientific and practical importance. The data presented are of more than national interest.
The introduction is presented correctly, in accordance with the subject. Numerous scientific articles, in concordance to the topic of the study, were consulted.
Methodology of the study was clearly presented, and appropriate to the proposed objectives.
The obtained results are important and have been analyzed and interpreted correctly, in accordance with the current methodology.
The discussions are appropriate, in the context of the results, and was conducted compared to other studies in the field.
The scientific literature, to which the reporting was made, is recent and representative in the field.
There are some minor changes I am suggesting in the detailed comments below.
Detailed comments
- Introduction
Na+ and Cl- - wrong format
Response: Well, the format has been corrected according to your suggestion.
ROS - The abbreviation “ROS" appears in the work for the first time - it should be explained.
Response: Thanks, the abbreviation “ROS” has been corrected to full form “Reactive oxygen species”.
2.3. Quantitative assays for PGP traits
FeCl3 - wrong format. Check and correct throughout the manuscript
Response: We have checked and corrected the format throughout the manuscript.
2.4. Soil collection, analysis and seed inoculation
(CFU) mL−1] - wrong format. Check and correct throughout the manuscript
Response: We have checked and corrected the format throughout the manuscript.
2.8. Radical scavenging capacity of Leaves
OD - The abbreviation “OD” appears in the work for the first time - it should be explained.
Response: Thanks, we corrected the abbreviation “OD” into full form “Optical density”.
Below the tables and figures, please add an explanation of what the lowercase letters in the columns mean
Response: Thank you for your comment. We have revised the MS and added an explanation of what the lowercase letters in the columns mean.
Figure 6 - Check and correct format
Response: Well, we have checked and revised the format of figure 6. Single blot/gel images have been rearranged in preparing figures. The fragments of the same original image were spliced together to re-order lanes and remove irrelevant lanes.
Reviewer 3 Report
Ali and colleagues present a study on induced salt tolerance in maize by PGPB Bacillus mycoides strain PM35. The presented data show a clear growth promotion in maize plants by B. mycoides under salt stress. Thus, the study is of interest to a broad readership of Life.
However, the manuscript requires a substantial revision before its publication. I would strongly recommend a substantial language revision to make important points clearer to the reader.
My main critic with the study is that the plant inoculation experiment was only performed once and it is not clear how many independent biological replicates per treatment were used. To confirm the results a second, independent experiment verifying the results is necessary in my point of view.
Moreover, the authors should elaborate the Materials and Methods part more in detail. For example the authors write: In each experimental unit, 20 ml of bacterial suspension was added. But what was the CFU per ml they used? Or The seeds were sown (6 surface sterilized seeds per pot) in plastic pots containing 200 g of sterilized soil. At this point is not clear to the reader what kind of seeds they used...origin, cultivar.
Author Response
Author's Response to Reviewer Comments
Dear Editor,
The authors gratefully acknowledge the reviewers for spending valuable time reviewing our manuscript and making constructive comments and suggestions. We believe that having followed the reviewer’s comments, the scientific and technical quality of the paper has been improved and fulfills the publication requirements of your esteemed journal. All the revisions are included by using the “track changes function” in the “manuscript with track changes file”
We are looking forward to hearing from you.
Yours faithfully
Baber Ali
(On behalf of Co-authors)
Reviewer 3:
Open Review
(x) I would not like to sign my review report
( ) I would like to sign my review report
English language and style
(x) Extensive editing of English language and style required
( ) Moderate English changes required
( ) English language and style are fine/minor spell check required
( ) I don't feel qualified to judge about the English language and style
|
|
|
Can be improved |
Must be improved |
Not applicable |
|
Does the introduction provide sufficient background and include all relevant references? |
( ) |
(x) |
( ) |
( ) |
|
Is the research design appropriate? |
( ) |
( ) |
(x) |
( ) |
|
Are the methods adequately described? |
( ) |
( ) |
(x) |
( ) |
|
Are the results clearly presented? |
( ) |
(x) |
( ) |
( ) |
|
Are the conclusions supported by the results? |
( ) |
(x) |
( ) |
( ) |
First of all, thank you so much for taking the time reviewing our manuscript. I am sure your considerations and corrections will give more reading comprehension and add more value to the paper. On behalf of the other authors, I appreciate the feedback from the Reviewer.
All suggestions made by the Reviewer were taken into account and the responses are highlighted with yellow color.
Comments and Suggestions for Authors
Ali and colleagues present a study on induced salt tolerance in maize by PGPB Bacillus mycoides strain PM35. The presented data show a clear growth promotion in maize plants by B. mycoides under salt stress. Thus, the study is of interest to a broad readership of Life.
However, the manuscript requires a substantial revision before its publication. I would strongly recommend a substantial language revision to make important points clearer to the reader.
My main critic with the study is that the plant inoculation experiment was only performed once and it is not clear how many independent biological replicates per treatment were used. To confirm the results a second, independent experiment verifying the results is necessary in my point of view.
Response: We understand the overall criticism and we have provided information about independent biological replicates per treatment used.
Moreover, the authors should elaborate the Materials and Methods part more in detail. For example the authors write: In each experimental unit, 20 ml of bacterial suspension was added. But what was the CFU per ml they used? Or The seeds were sown (6 surface sterilized seeds per pot) in plastic pots containing 200 g of sterilized soil. At this point is not clear to the reader what kind of seeds they used...origin, cultivar.
Response: Thank you for your feedback to improve our MS. As per your kind feedback, we have added the information of CFU per ml, we used in our experiment. The information of seeds variety in the MS (Maize variety named SG-2002) has been added.
Reviewer 4 Report
The manuscript life-1489246 reports interesting results on the salinity tolerance effects induced in maize by Bacillus mycoides PM35 inoculation. The strain was first investigated in vitro for salinity tolerance and plant growth-promoting traits (indole acetic acid, siderophore, ACC-deaminase, and exopolysaccharides). Subsequently, B. mycoides PM35 salinity tolerance induction was investigated in maize under gnotobiotic conditions.
The authors used valid methodologies, the data handling was appropriate and manuscript organization is satisfactory. However, I was not able to access the supplementary material within the system and there are some aspects related to the Introduction section and quality of the presentation that should be addressed. Please, see specific comments.
Introduction
The introduction places the study in a broad context and underlines why the study is important. However, the Bacillus mycoides should be introduced better and a clear working hypothesis should be introduced.
- Be consistent with PGPB/PGPR/PGP bacteria/PGP rhizobacteria use. I would introduce PGPB concept and be consistent with this abbreviation all over the manuscript.
- For statements in lines "Therefore, PGPR, producing phytohormones, proline....". Are these mechanisms described for your strain? I would provide after these lines the working hypothesis statement.
Materials and methods
The details provided for the method are sufficient to replicate the tests. I suggest polishing the English language in some parts:
- "Guimarães et al. [27] used a microtiter plate-based technique with some slight adjustments to quantify biofilm formation." Is this statement means that you used the method described by Guimarães et al.?
- "Bacillus mycoides PM35 bacteria were cultured on LB medium amended with vari-ous levels of salinity stress [24]." Please revise with "B. mycoides PM35 was cultured on LB medium amended with various concentrations of salinity stress [24]."
Results
The results organization and presentation is clear. If possible the quality of figures 1, 2, 3, 5, and 7 should be improved.
I suggest authors calculate and introduce in their dataset the chlorophyll a and b ratio. This ratio is an important indicator of plant response to stresses, particularly salt stress.
Discussion
The authors correctly discussed the findings in relation to the working hypothesis and previous studies.
Conclusions
This section is appropriate as the manuscript contains many elements. However, the following statements must be revised:
- "This investigation highlights the effectiveness of PM35 to serve as bio-inoculant for mitigation of salinity stress in maize plants and could be used as eco-friendly viable option for plant growth and yield production under saline environment." This statement is not supported by results. The authors performed only an experiment in gnotobiotic conditions. Before declaring the previous speculations, the experiment should be carried out in greenhouse and in open field experiments and under different climatic conditions. I would remove these statements and provide more details about future studies that would help to get a confirmation of the findings obtained
- Revise "Bacillus sp. PM35" with "B. mycoides PM35"
Other comments
After the first time used in the main text, Bacillus mycoides PM35 name could be abbreviated as B. mycoides PM35.
Revise italics throughout the manuscript.
Revise the English language for grammar errors, typos and clarity.
Author Response
Author's Response to Reviewer Comments
Dear Editor,
The authors gratefully acknowledge the reviewers for spending valuable time reviewing our manuscript and making constructive comments and suggestions. We believe that having followed the reviewer’s comments, the scientific and technical quality of the paper has been improved and fulfills the publication requirements of your esteemed journal. All the revisions are included by using the “track changes function” in the “manuscript with track changes file”
We are looking forward to hearing from you.
Yours faithfully
Baber Ali
(On behalf of Co-authors)
Reviewer 4:
Open Review
(x) I would not like to sign my review report
( ) I would like to sign my review report
English language and style
( ) Extensive editing of English language and style required
(x) Moderate English changes required
( ) English language and style are fine/minor spell check required
( ) I don't feel qualified to judge about the English language and style
|
|
|
Can be improved |
Must be improved |
Not applicable |
|
Does the introduction provide sufficient background and include all relevant references? |
( ) |
(x) |
( ) |
( ) |
|
Is the research design appropriate? |
(x) |
( ) |
( ) |
( ) |
|
Are the methods adequately described? |
(x) |
( ) |
( ) |
( ) |
|
Are the results clearly presented? |
(x) |
( ) |
( ) |
( ) |
|
Are the conclusions supported by the results? |
( ) |
(x) |
( ) |
( ) |
First of all, thank you so much for taking the time reviewing our manuscript. I am sure your considerations and corrections will give more reading comprehension and add more value to the paper. On behalf of the other authors, I appreciate the feedback from the Reviewer.
All suggestions made by the Reviewer were taken into account and the responses are highlighted with yellow color.
Comments and Suggestions for Authors
The manuscript life-1489246 reports interesting results on the salinity tolerance effects induced in maize by Bacillus mycoides PM35 inoculation. The strain was first investigated in vitro for salinity tolerance and plant growth-promoting traits (indole acetic acid, siderophore, ACC-deaminase, and exopolysaccharides). Subsequently, B. mycoides PM35 salinity tolerance induction was investigated in maize under gnotobiotic conditions.
The authors used valid methodologies, the data handling was appropriate and manuscript organization is satisfactory. However, I was not able to access the supplementary material within the system and there are some aspects related to the Introduction section and quality of the presentation that should be addressed. Please, see specific comments.
Introduction
The introduction places the study in a broad context and underlines why the study is important. However, the Bacillus mycoides should be introduced better and a clear working hypothesis should be introduced.
Response: Well, thank you for your feedback. We have revised the introduction and added a clear hypothesis and more information about B. mycoides PM35 to introduce better in the MS (Last paragraph of introduction).
Be consistent with PGPB/PGPR/PGP bacteria/PGP rhizobacteria use. I would introduce the PGPB concept and be consistent with this abbreviation all over the manuscript.
Response: Thanks. We have revised the whole MS and introduced the term “PGPB” as your recommendation in the whole MS.
For statements in lines "Therefore, PGPR, producing phytohormones, proline....” Are these mechanisms described for your strain? I would provide after these lines the working hypothesis statement.
Response: Thank you. We revised the MS and provided a hypothesis statement in the last paragraph of the introduction after these lines.
Materials and methods
The details provided for the method are sufficient to replicate the tests. I suggest polishing the English language in some parts:
"Guimarães et al. [27] used a microtiter plate-based technique with some slight adjustments to quantify biofilm formation." Is this statement means that you used the method described by Guimarães et al.?
Response: The sentence was rewritten to improve the sentence structure and for more clarity.
"Bacillus mycoides PM35 bacteria were cultured on LB medium amended with vari-ous levels of salinity stress [24]." Please revise with "B. mycoides PM35 was cultured on LB medium amended with various concentrations of salinity stress [24]."
Response: We improved the sentence and revised it as “B. mycoides PM35 was cultured on LB medium amended with various concentrations of salinity stress [24]."
Results
The results organization and presentation is clear. If possible the quality of figures 1, 2, 3, 5, and 7 should be improved.
Response: All the figures were updated and changes were made in the image quality, font size, image size, and graph distribution
I suggest authors calculate and introduce in their dataset the chlorophyll a and b ratio. This ratio is an important indicator of plant response to stresses, particularly salt stress.
Response: The suggestion is highly appreciated but we have already not only analyzed chlorophyll a, b, and total chlorophyll but all the parameters that are affected by salinity stress into the Co-efficient correlation (Figure 9) which fully explains the response of the plant to salt stress. I think this analysis covers the suggested matter.
Discussion
The authors correctly discussed the findings in relation to the working hypothesis and previous studies.
Conclusions
This section is appropriate as the manuscript contains many elements. However, the following statements must be revised:
"This investigation highlights the effectiveness of PM35 to serve as bio-inoculant for mitigation of salinity stress in maize plants and could be used as eco-friendly viable option for plant growth and yield production under saline environment." This statement is not supported by results. The authors performed only an experiment in gnotobiotic conditions. Before declaring the previous speculations, the experiment should be carried out in greenhouse and in open field experiments and under different climatic conditions. I would remove these statements and provide more details about future studies that would help to get a confirmation of the findings obtained
Response: We have extensively modified the entire discussion section providing a much more focused discussion
Revise "Bacillus sp. PM35" with "B. mycoides PM35"
Response: We have revised the whole MS with "B. mycoides PM35".
Other comments
After the first time used in the main text, Bacillus mycoides PM35 name could be abbreviated as B. mycoides PM35.
Response: We have revised the whole MS with "B. mycoides PM35".
Revise italics throughout the manuscript.
Response: We have revised italics throughout the manuscript.
Revise the English language for grammar errors, typos and clarity.
Response: The whole manuscript was revised for typing and English grammar mistakes.
Round 2
Reviewer 3 Report
The authors inoculated only three plants per treatment with a concentration of 10^8 to 10^9 cfu per ml. First, three plants is the minimum for any statistical analysis. Secondly, there was no uniform amount of bacteria applied to the plants. Between 10^8 cfu per ml and 10^9 cfu per ml is a huge difference.
I am sorry but I don´t feel that the manuscript is publishable in its current form and a second experiment with a defined amount of bacteria like 10^8 cfu ml and at least 5-6 replicates for the plant experiment is needed.
In addition, the authors responded to reviewer 4: The whole manuscript was revised for typing and English grammar mistakes. I have to say that this is not the case. Please take a look at the reference list, there are still many mistakes.
Author Response
Author's Response to Reviewer Comments
Dear Editor,
The authors gratefully acknowledge the reviewers for spending valuable time reviewing our manuscript and making constructive comments and suggestions. We believe that having followed the reviewer’s comments, the scientific and technical quality of the paper has been improved and fulfills the publication requirements of your esteemed journal. All the revisions are included by using the “track changes function” in the “manuscript with track changes file”
We are looking forward to hearing from you.
Yours faithfully
Baber Ali
(On behalf of Co-authors)
Reviewer 3:
Open Review
(x) I would not like to sign my review report
( ) I would like to sign my review report
English language and style
(x) Extensive editing of English language and style required
( ) Moderate English changes required
( ) English language and style are fine/minor spell check required
( ) I don't feel qualified to judge about the English language and style
|
|
|
Can be improved |
Must be improved |
Not applicable |
|
Does the introduction provide sufficient background and include all relevant references? |
( ) |
(x) |
( ) |
( ) |
|
Is the research design appropriate? |
( ) |
( ) |
(x) |
( ) |
|
Are the methods adequately described? |
( ) |
( ) |
(x) |
( ) |
|
Are the results clearly presented? |
( ) |
(x) |
( ) |
( ) |
|
Are the conclusions supported by the results? |
( ) |
(x) |
( ) |
( ) |
First of all, thank you so much for taking the time to review our manuscript. I am sure your considerations and corrections will give more reading comprehension and add more value to the paper. On behalf of the other authors, I appreciate the feedback from the Reviewer.
All suggestions made by the Reviewer were taken into account and the responses are highlighted with yellow color.
Comments and Suggestions for Authors
The authors inoculated only three plants per treatment with a concentration of 10^8 to 10^9 CFU per ml. First, three plants are the minimum for any statistical analysis. Secondly, there were no uniform amount of bacteria applied to the plants. Between 10^8 CFU per ml and 10^9 CFU per ml is a huge difference.
Response.
Thanks for your concern and we really appreciate it. The minimum replications for any statistical analysis are three and the reviewer also mentioned it and we have three replicates in each treatment and each replicate (pot) has six plants. (1*6=6, 3*6=18). This is how each treatment encompasses 18 plants. We plainly elaborated it into the M&M. Page, 10 line 222
These papers have been published with three replicates from our lab in 2022 and 2019 and the results are reproducible. Journal of plant growth regulation, Plant Physiology and Biochemistry
https://link.springer.com/article/10.1007/s00344-021-10571 4?utm_source=xmol&utm_medium=affiliate&utm_content=meta&utm_campaign=DDCN_1_GL01_metadata
https://www.sciencedirect.com/science/article/pii/S0981942819301317?casa_token=7f06A5IpMi4AAAAA:uyWrlpQUpblZgvkZLHCoKrCvESZ3ICTzfV9vfd9S9Amv9AuRzEWEaz8pQ_67JIni337NUASmSQ
Regarding the concentration of bacterial suspension, the reviewer is right 10^8 per ml and we applied 10^8 bacterial suspension in each treatment but mistakenly, we have written 10^8 to 10^9 into the M&M and unfortunately, we suppose to correct it in the first revision but we couldn’t. Now it has been corrected according to the reviewer suggestion
I am sorry but I don´t feel that the manuscript is publishable in its current form and a second experiment with a defined amount of bacteria like 10^8 CFU ml and at least 5-6 replicates for the plant experiment is needed.
Response.
Recently the following articles have been published from our lab where we carried out our all experiments with one trial and containing three replicates. The protocols were optimized and got reproducible results. We really appreciate the productive suggestions of the reviewer and we will maintain six replicates in future experiments.
Agronomy. Sustainability. MDPI, Journal of plant growth regulation
https://www.mdpi.com/2073-4395/11/10/1907
https://www.mdpi.com/2073-4395/11/9/1820/htm
https://www.mdpi.com/2071-1050/13/24/13529/htm
https://link.springer.com/article/10.1007/s00344-021-10571 4?utm_source=xmol&utm_medium=affiliate&utm_content=meta&utm_campaign=DDCN_1_GL01_metadata
In addition, the authors responded to reviewer 4: The whole manuscript was revised for typing and English grammar mistakes. I have to say that this is not the case. Please take a look at the reference list, there are still many mistakes.
Response.
Thank you. We checked MS and all the references carefully. MS has been revised for English, grammar, typing mistakes and corrected. All the references have been updated and properly cited at their corresponding positions. All the references have been cited by using “Mendeley software” to resolve the matter of incomplete references.
Reviewer 4 Report
The authors correctly addressed my previous suggestions and responded to all my comments appropriately. I have no further suggestions.
Author Response
We truly appreciate the reviewer who review our paper and made suggestions and recommendations that improved the quality of the research article. Thank you for your feedback.